CALT-TH 2021-017

# Navigator Function for the Conformal Bootstrap

**Marten Reehorst**[a]**, Slava Rychkov**[a,b]**, David Simmons-Duffin**[c]**,**

**Benoit Sirois**[b,a]**, Ning Su**[d]**, Balt van Rees**[e]

[a] Institut des Hautes Études Scientifiques, 91440 Bures-sur-Yvette, France
[b] Laboratoire de Physique de l'Ecole normale supérieure, ENS,
Université PSL, CNRS, Sorbonne Université, Université de Paris, F-75005 Paris, France
[c] Walter Burke Institute for Theoretical Physics, Caltech, Pasadena, CA 91125, USA
[d] Department of Physics, University of Pisa, I-56127 Pisa, Italy
[e] CPHT, CNRS, École Polytechnique, Institut Polytechnique de Paris,
Route de Saclay, 91128 Palaiseau, France

## Abstract

Current numerical conformal bootstrap techniques carve out islands in theory space by repeatedly checking whether points are allowed or excluded. We propose a new method for searching theory space that replaces the binary information "allowed"/"excluded" with a continuous "navigator" function that is negative in the allowed region and positive in the excluded region. Such a navigator function allows one to efficiently explore high-dimensional parameter spaces and smoothly sail towards any islands they may contain. The specific functions we introduce have several attractive features: they are well-defined in large regions of parameter space, can be computed with standard methods, and evaluation of their gradient is immediate due to an SDP gradient formula that we provide. The latter property allows for the use of efficient quasi-Newton optimization methods, which we illustrate by navigating towards the 3d Ising island.

April 2021

# 1 Introduction and summary

Over the last decade, the numerical conformal bootstrap program[1] has relied on the idea [4] that for any point in CFT parameter space it is possible to check if the point is allowed or excluded by constructing positive linear functionals. In this work we will dramatically upgrade this idea, replacing the binary information "allowed/excluded" by a continuous measure of success, called a "navigator function." For excluded points, the navigator function will tell us how far we are from the allowed region. Minimizing the navigator, we will be able to quickly find the allowed region, starting from an excluded point. For allowed points, the navigator will tell us how far inside the allowed region we are, and navigator minima will be excellent predictors for the position of an actual CFT.

To describe what we have in mind in some detail, let $X$ be an infinite-dimensional vector containing all parameters characterizing a CFT (i.e. all operator dimensions and OPE coefficients, bundled together). We split it as $X = (x, y)$ where $x \in \mathbb{R}^k$ are parameters we are especially interested in, and $y$ contains all the rest. We also select a finite subset of the infinitely many bootstrap equations.

Most bootstrap computations performed so far proceeded in what one may call "oracle mode."[2] One picks a sequence of trial vectors $x_1, x_2, \ldots$ and asks for each of them if there is any $y$ such that $X = (x_i, y)$ satisfies the selected subset of bootstrap equations. A bootstrap solver such as SDPB [5, 6] provides an answer: "allowed" or "excluded". By trying many $x_i$'s, one maps out the allowed region.[3] Thus, we compute the characteristic function $\chi_R$ of the allowed region $R$ (i.e. $\chi_R(x) = 1$ for $x \in R$ and $\chi_R(x) = 0$ otherwise). Experience shows that the boundary of the allowed region $\partial R$ is typically smooth, apart from isolated points (kinks). This can guide the choice of future trial points and speed up the computation.[4] By trying many points, one zooms in on the boundary $\partial R$ of the allowed region. Importantly, a single oracle query does not provide any information about whether one is close to or far from $\partial R$. Rather, one knows that one is close to $\partial R$ if one can find two nearby trial points $x_i$ and $x_{i'}$ such that they are on two different sides of the boundary.

We will modify this setup so that a single SDPB run computes a continuous function $\mathcal{N}(x)$, called a *navigator*, which will give a more nuanced measure of success than simply "allowed/excluded." To be maximally useful, the navigator should have the following properties:

- $\mathcal{N}(x)$ is continuous and differentiable;

---

[1]Ssee [1] for a thorough review, and [2, 3] for pedagogical introductions.

[2]In technical jargon referred to as "feasibility mode."

[3]Other typical bootstrap computations are OPE coefficient optimizations. Sometimes these computations allow to zoom in on actual CFTs, as e.g. $c$-minimization is conjectured to lead to the 3d Ising CFT [7].

[4]Other speed-up tricks include the cutting surface algorithm [8], which allows in some cases to use a single oracle computation to rule out not just one point but a large swath of the parameter space.

- $\mathcal{N}(x) > 0$ outside the allowed region $R$, and $\mathcal{N}(x) < 0$ inside $R$. In particular, $\mathcal{N}(x) = 0$ on the boundary $\partial R$;[5]

- $\mathcal{N}(x)$ should be defined not just in a tiny neighborhood of the allowed region but globally;

- The allowed region $R$ should be a basin of attraction of the navigator function from a sizable neighborhood of $R$.

Assuming these nice properties, the navigator value will allow us to guess how far we are from the allowed region. We will also be able to reach the allowed region by starting from some initial trial point $x_0$ and by minimizing the navigator until we reach a point with negative $\mathcal{N}(x)$. We'd like to be optimistic and hope that the navigator has no local minima away from the allowed region where such a search may get stuck.

The idea of replacing the binary information of "oracle mode" with continuous information from solving an optimization problem is not completely new [9, 10]. Notably Ref. [10] emphasized the power of this idea to quickly determine the boundary of the allowed region once its approximate position is known, replacing bisection with the secant method.[6] A crucial difference here is our requirement that the navigator should be defined in a wide region and not only near the boundary, which will greatly increase the list of potential applications. This requirement is non-trivial and the early navigator avatars [9, 10] don't satisfy it (see Section 2.1.3).

In this paper we will lay down the systematic theory of navigator functions by showing three important results:

1. First, we will show that navigators satisfying all of the above properties can indeed be found for a generic bootstrap problem. We will present both the general principle of their existence, and several explicit constructions (see Section 2). Please scroll down to Fig. 1 for a concrete navigator example in the mixed $\sigma$-$\epsilon$ bootstrap setup used to isolate the 3d Ising model. It has all the nice properties, and in particular a single minimum (within the range we show), located within the 3d Ising island. See Section 3 for more beautiful navigator plots.

2. Our navigators can be evaluated using standard conformal bootstrap software such as SDPB. In practical applications that we have in mind, it's important to know not just the navigator but also its gradient. Our second important result is a general "SDP gradient formula," Eq. (4.16). This formula shows that navigator gradient can be evaluated essentially for free once the navigator value has been computed using SDPB.

---

[5]In the Level Set Method of computational geometry, such functions are called "level set functions" or "level set fields". Closely related are also "boundary defining functions" of differential geometry, which however are only required to be defined near the boundary.

[6]We will see below that the navigator derivative can be evaluated "for free," allowing to replace the secant method with the even faster Newton method.

3. We foresee that one of the most important navigator applications will be to quickly look for allowed points, i.e. to "sail towards the Ising island," by minimizing the navigator. Naive minimization strategies, such as the gradient descent, are inefficient, getting stuck in narrow "valleys" of the navigator surface. Our third important result is to demonstrate how a quasi-Newton method—the BFGS algorithm [11]—successfully overcomes these difficulties (Section 5). This algorithm finds first the allowed region, and then the navigator minimum, in a relatively small number of steps.

The paper is structured as follows. Section 2 will explain our two main navigator constructions: the GFF-navigator and the $\Sigma$-navigator. (A third construction is in App. B). In Section 3 we will show various plots of these navigators, to gain intuition about their shape. In Section 4 we will derive the SDP gradient formula. In Section 5 we will describe the BFGS algorithm and its bounding-box modification, to look for an allowed point and the navigator minimum, and show that it performs well in realistic multiple-correlator setups. In Section 6 we describe another possible navigator application: extremizing operator dimension within the allowed region. This represents an attractive alternative to the `tiptop` algorithm recently introduced for this purpose in the feasibility setup [12]. In Section 7 we conclude. Appendix C shows how one can also evaluate the navigator Hessian, in addition to the gradient, provides numerical tests of these procedures.

# 2 Navigator function

Our motivation to look for the navigator function, and its desired properties, have already been described in the introduction. The crucial requirement is that the navigator should be *finite*. Indeed, a navigator which is negative inside the allowed region and equals $+\infty$ outside would be rather useless for the purposes we have in mind, such as looking for an allowed point starting from an excluded one. Furthermore, once a finite navigator is constructed, other nice properties turn out to also be satisfied.

How to get a robustly finite navigator is one of the main ideas of our paper (see Section 2.1.3 for an account of naive attempts which fail). Although the idea is general, we will start in Section 2.1 by presenting it in the simplest single-correlator setup. We will then move on to more realistic multiple-correlator problems.

## 2.1 Single-correlator problems

Consider the simplest bootstrap setup: scalar gap maximization in a single 4pt function of four identical scalars [4]. Thus we are solving the bootstrap equation

$$F_{0,0}(u,v) + \sum_{(\Delta,\ell) \in S(\Delta_*)} p_{\Delta,\ell} F_{\Delta,\ell}(u,v) = 0, \qquad p_{\Delta,\ell} \geqslant 0 \qquad (2.1)$$

where $F_{\Delta,\ell}(u,v) = v^{\Delta_\phi} g_{\Delta,\ell}(u,v) - u^{\Delta_\phi} g_{\Delta,\ell}(v,u)$. Here $\Delta_\phi$ is the external scalar dimension which for simplicity is considered fixed (although see footnote 7). The set $S(\Delta_*)$ is given by:

$$S(\Delta_*) = \{(\Delta,\ell) : \ell = 0 \text{ and } \Delta \geqslant \Delta_*, \text{ or } \ell = 2,4,\ldots \text{ and } \Delta \geqslant \ell + d - 2\}. \qquad (2.2)$$

The variables to be solved for in (2.1) are the set of appearing pairs $(\Delta,\ell)$ and the corresponding coefficients $p_{\Delta,\ell}$. We are interested to know what is the maximal $\Delta_*$ such that (2.1) has a solution.

We would like to define a navigator function $\mathcal{N}(\Delta_*)$ such that it is negative if a solution exists and is positive if it does not exist. To this end we will consider a modified problem of the form

$$F_{0,0}(u,v) + \lambda M(u,v) + \sum_{(\Delta,\ell)\in S(\Delta_*)} p_{\Delta,\ell} F_{\Delta,\ell}(u,v) = 0, \qquad p_{\Delta,\ell} \geqslant 0, \qquad (2.3)$$

We just added an extra term in the l.h.s. with a fixed function $M(u,v)$ and a new parameter $\lambda$. The function $M(u,v)$ will be chosen so that the following crucial property holds:

★ For any $\Delta_*$, problem (2.3) has a solution with *some* $\lambda = \lambda_0(\Delta_*) > 0$. (2.4)

Given this property, the navigator function will be defined as the *minimal* value of $\lambda$ such that (2.3) has a solution:[7]

$$\mathcal{N}(\Delta_*) = \min \lambda \text{ such that } (2.3) \text{ has a solution.} \qquad (2.5)$$

Property (2.4) then guarantees that the navigator is bounded from above, as we have $\mathcal{N}(\Delta_*) \leqslant \lambda_0(\Delta_*)$. We also see that the navigator is monotonically non-decreasing in the $\Delta_*$ direction, negative in the allowed region and positive outside.[8]

This described construction does not formally guarantee other nice properties of the navigator that we wish to have (that $\mathcal{N}(\Delta_*)$ is differentiable, strictly negative in the allowed region, has no local minima outside the allowed region where minimization can get stuck etc.) It also does not guarantee that the navigator is finite inside the allowed region (it may be $-\infty$ there). Nevertheless, explicit navigator functions constructed below using this idea will have all these additional nice properties, by inspection.

We will now give two examples of functions $M(u,v)$ that have the required property (2.4).

---

[7]Although in this section we consider $\Delta_\phi$ fixed, it is trivial to relax this and consider the navigator as a function of both $\Delta_\phi$ and $\Delta_*$, defined by the same Eq. (2.5). The zero set of $\mathcal{N}(\Delta_\phi, \Delta_*)$ is then a curve which is the upper bound on $\Delta_*$ as a function of $\Delta_\phi$. We will not develop this idea further here but we will encounter analogous situations below in the multiple-correlator context.

[8]Note that for any $\Delta_*$ the set of $\lambda$'s for which (2.3) has a solution is a connected subset of the real axis. This follows from the fact that a convex linear combination of solutions is again a solution.

### 2.1.1 GFF-navigator

We know that for any $\Delta_\phi$, Eq. (2.1) has a Generalized Free Field (GFF) solution with the spectrum $\Delta = 2\Delta_\phi + 2n + \ell$, $n \geqslant 0$, $\ell = 0, 2, 4, \ldots$, corresponding to operators of schematic form $\phi \partial^\ell \Box^n \phi$. The GFF-navigator is obtained by taking $M(u, v)$ to be the first term in this solution:

$$M_{\mathrm{GFF}}(u, v) = 2F_{2\Delta_\phi, 0}(u, v). \tag{2.6}$$

Here 2 is the square of the GFF OPE coefficient in the OPE $\phi \times \phi \ni \sqrt{2}\mathcal{O}$, where $\mathcal{O} = \frac{1}{\sqrt{2}}\phi^2$ is unit-normalized. The GFF solution to crossing provides a solution to (2.3) with $\lambda = 1$ as long as all GFF operators besides $\phi^2$ belong to $S(\Delta_*)$, which will be the case for $\Delta_* \leqslant 2\Delta_\phi + 2$. Hence $\mathcal{N}(\Delta_*) \leqslant 1$ for any $\Delta_*$ in this range.

Note that having a finite navigator in the range $\Delta_* \leqslant 2\Delta_\phi + 2$ is sufficient for the problem at hand, since the boundary of the allowed region for (2.1) is known to satisfy this condition. Alternatively, higher GFF operators which do not satisfy gap assumptions may be added to the r.h.s. of Eq. (2.6). See App. A for this tweak of the GFF-navigator, important for bootstrap problems with additional gaps in the spectrum.

### 2.1.2 $\Sigma$-navigator

Another possibility, called the $\Sigma$-navigator, results from choosing:

$$M_\Sigma(u, v) = -\sum_{i=1}^{n} c_i F_{\Delta_i, \ell_i}(u, v), \tag{2.7}$$

where $(\Delta_i, \ell_i)$ are any $n$ spectrum points in $S(\Delta_*)$, $c_i > 0$ some fixed positive coefficients, and $n$ is a sufficiently large number. Since the coefficients $c_i$ are, apart from being positive, essentially arbitrary, there is a lot of freedom in choosing the $\Sigma$-navigator.

Consider Eq. (2.3) with this $M(u, v)$. In practice, in the numerical conformal bootstrap we analyze this equation in Taylor expansion around some point, i.e. we replace functions of $u, v$ by vectors of Taylor coefficients of some finite length $n_0$. Denoting vectors by boldface symbols, we have

$$\mathbf{F}_{0,0} + \lambda \mathbf{M}_\Sigma + \sum_{(\Delta, \ell) \in S(\Delta_*)} p_{\Delta, \ell} \mathbf{F}_{\Delta, \ell} = 0, \qquad p_{\Delta, \ell} \geqslant 0. \tag{2.8}$$

We claim that this equation will generically have a solution with some positive $\lambda$ as long as the number of terms $n$ in (2.7) is $n \geqslant n_0$. Indeed, generically the vectors $\mathbf{F}_{\Delta_i, \ell_i}$ are not expected to be linearly independent. Thus the equation

$$\mathbf{F}_{0,0} + \sum_{i=1}^{n} x_i \mathbf{F}_{\Delta_i, \ell_i} = 0, \tag{2.9}$$

will have a solution as longs as $x_i$ are allowed to have either sign. We rewrite this solution as

$$\mathbf{F}_{0,0} + \lambda \mathbf{M}_\Sigma + \sum_{i=1}^{n}(x_i + \lambda c_i)\mathbf{F}_{\Delta_i,\ell_i} = 0, \tag{2.10}$$

For sufficiently large positive $\lambda = \lambda_0$ all the coefficients $x_i + \lambda_0 c_i \geqslant 0$ so this is a solution to (2.8), proving the above claim. Hence, by the general arguments, the navigator is bounded from above by $\lambda_0$.

In the described construction the number of terms $n$ in (2.7) may have to be increased with the number of conformal block derivatives used in the numerical analysis. Alternatively, we may replace the sum in (2.7) by an integral with a positive continuous measure in some interval of $\Delta$'s. Then the same navigator may be used independently of the number of derivatives.

### 2.1.3 Dual picture

In the dual approach to the numerical conformal bootstrap, the problem of computing the navigator (2.5) is formulated as follows:

$$\mathcal{N}(\Delta_*) = \max \alpha(F_{0,0}) \text{ over all linear functionals } \alpha \text{ such that}$$
$$\alpha(M) = -1$$
$$\alpha(F_{\Delta,\ell}) \geqslant 0 \text{ for all } (\Delta,\ell) \in S(\Delta_*) \tag{2.11}$$

Our construction guarantees that the choices (2.6) or (2.7) lead to this problem having a solution bounded from above for any $\Delta_*$.

From this dual formulation we can see that the $\Sigma$-navigator is guaranteed to be finite also in the allowed region (i.e. it cannot be $-\infty$ there). That's because for any $\Delta_*$ there is always some functional which satisfies the positivity condition in (2.11). Rescaling this functional we may make it also satisfy the normalization condition. This provides a finite lower bound for the $\Sigma$-navigator. For the GFF-navigator this argument clearly fails if $\Delta_* \leqslant 2\Delta_\phi$. In this case there is no functional $\alpha$ satisfying both the normalization and the positivity conditions. Thus the GFF-navigator equals $-\infty$ for $\Delta_* \leqslant 2\Delta_\phi$.[9] This is not so problematic in practice, since this range is anyway deep inside the allowed region for the single-correlator problem. In principle the GFF-navigator could become $-\infty$ even for $\Delta_*$ somewhat above $2\Delta_\phi$, but we have not seen this happen.

It is instructive to compare the above dual formulation with how one computes the maximal allowed value $p^{\max}_{\Delta_0,\ell_0}$ of the squared OPE coefficient for an operator $(\Delta_0, \ell_0)$ present in the spectrum [13, 9]:

$$p^{\max}_{\Delta_0,\ell_0} = -\max \alpha(F_{0,0}) \text{ over all linear functionals } \alpha \text{ such that}$$
$$\alpha(F_{\Delta_0,\ell_0}) = 1$$
$$\alpha(F_{\Delta,\ell}) \geqslant 0 \text{ for all } (\Delta,\ell) \in S(\Delta_*) \tag{2.12}$$

---

[9]This is also obvious from the primal definition (2.6).

Comparing (2.12) with (2.11), one may wonder if one could perhaps define a navigator simpler than in our proposals, namely as

$$\mathcal{N}(\Delta_*) = -p^{\max}_{\Delta_0,\ell_0} \qquad (?) \tag{2.13}$$

for some appropriate choice of $(\Delta_0, \ell_0)$ in $S(\Delta_*)$. E.g. what if one tries $\ell_0 = 0$ and $\Delta_0$ a little above the boundary of the allowed region? It turns out however that such simple-minded choices of functional normalization are inadequate. Namely, they give a finite navigator only in a rather small neighborhood of the boundary of the allowed region, which moreover gets smaller and smaller as one increases the number of derivatives used in the conformal bootstrap computation.[10] If one already knows quite well where the boundary is (e.g. via bisection), then using this navigator one can quickly determine it even more precisely. But if one starts far away from the boundary, this navigator would not help. Our $\Sigma$-navigator proposal shows that to get a robustly bounded navigator one needs to modify this idea by normalizing not on a single conformal block in the allowed region as in (2.12) but on a positive linear combination of many blocks as in (2.7).

Analogously, one could have hoped to get a bounded navigator by normalizing the functional to $-1$ on a single conformal block in the region outside $S(\Delta_*)$. But again, one finds that choosing $\ell_0 = 0$ and $\Delta_0$ a little below the boundary of the allowed region gives a navigator which is finite only in a small neighborhood of the boundary of the allowed region. Instead, our GFF-navigator proposal shows that if $\Delta_0$ is lowered all the way to $2\Delta_\phi$, which is quite a bit lower than the boundary of the allowed region, then the navigator becomes robustly bounded from above.

## 2.2 Multiple-correlator problems

We will now discuss how the navigator function construction generalizes to bootstrap problems involving several correlation functions. The main idea will be the same: we just need to add a new term so that crossing can always be obeyed, and minimize its coefficient.

We will consider the example of three 4pt functions $\langle \sigma\sigma\sigma\sigma \rangle$, $\langle \sigma\sigma\epsilon\epsilon \rangle$ and $\langle \epsilon\epsilon\epsilon\epsilon \rangle$ where $\sigma$ and $\epsilon$ are an odd and even scalars in a $\mathbb{Z}_2$-invariant CFT (such as the critical 3d Ising model). This system of correlators leads to 5 independent crossing relations [14]:

$$\sum_{\mathcal{O}^+} \text{Tr}\left[ P_{\mathcal{O}} \vec{V}_{+,\Delta,\ell} \right] + \sum_{\mathcal{O}^-} p_{\mathcal{O}} \vec{V}_{-,\Delta,\ell} = 0, \tag{2.14}$$

$$P_{\mathcal{O}} = \begin{pmatrix} \lambda_{\sigma\sigma\mathcal{O}} & \lambda_{\epsilon\epsilon\mathcal{O}} \end{pmatrix} \otimes \begin{pmatrix} \lambda_{\sigma\sigma\mathcal{O}} \\ \lambda_{\epsilon\epsilon\mathcal{O}} \end{pmatrix}, \qquad p_{\mathcal{O}} = \lambda^2_{\sigma\epsilon\mathcal{O}}, \tag{2.15}$$

---

[10]Ref. [10] considered an early version of navigator function corresponding to normalizing one particular component of the functional to 1. This navigator prototype suffered from the same problem of being finite only in a small region. We are grateful to Tom Hartman and Amir Tajdini for enlightening communications concerning their findings, which sparked our search for a robust navigator function.

where $\vec{V}_{-,\Delta,\ell}$ is a 5-vector of functions while $\vec{V}_{+,\Delta,\ell}$ is a 5-vector of $2 \times 2$ symmetric matrices of functions of $u, v$:

$$\vec{V}_{+,\Delta,\ell} = \begin{pmatrix} \begin{pmatrix} F_{-,\Delta,\ell}^{\sigma\sigma,\sigma\sigma} & 0 \\ 0 & 0 \end{pmatrix} \\ \begin{pmatrix} 0 & 0 \\ 0 & F_{-,\Delta,\ell}^{\epsilon\epsilon,\epsilon\epsilon} \end{pmatrix} \\ \begin{pmatrix} 0 & 0 \\ 0 & 0 \end{pmatrix} \\ \begin{pmatrix} 0 & \frac{1}{2}F_{-,\Delta,\ell}^{\sigma\sigma,\epsilon\epsilon} \\ \frac{1}{2}F_{-,\Delta,\ell}^{\sigma\sigma,\epsilon\epsilon} & 0 \end{pmatrix} \\ \begin{pmatrix} 0 & \frac{1}{2}F_{+,\Delta,\ell}^{\sigma\sigma,\epsilon\epsilon} \\ \frac{1}{2}F_{+,\Delta,\ell}^{\sigma\sigma,\epsilon\epsilon} & 0 \end{pmatrix} \end{pmatrix}, \qquad \vec{V}_{-,\Delta,\ell} = \begin{pmatrix} 0 \\ 0 \\ F_{-,\Delta,\ell}^{\sigma\epsilon,\sigma\epsilon} \\ (-1)^{\ell}F_{-,\Delta,\ell}^{\epsilon\sigma,\sigma\epsilon} \\ -(-1)^{\ell}F_{+,\Delta,\ell}^{\epsilon\sigma,\sigma\epsilon} \end{pmatrix}. \tag{2.16}$$

See [14] for the expressions of the functions $F_{\pm,\Delta,\ell}^{ij,kl}(u,v)$. The first sum in (2.14) runs over the $\mathbb{Z}_2$-even operators $\mathcal{O}^+$ in the OPEs $\sigma \times \sigma$ and $\epsilon \times \epsilon$ (whose spin is necessarily even), while the second sum in (2.14) is over all $\mathbb{Z}_2$-odd operators $\mathcal{O}^-$ in the OPE $\sigma \times \epsilon$ (which can have any spin).

As usual, we will treat separately the unit operator contribution

$$\vec{V}_{0,0} = \text{Tr}\left[P_{0,0}\vec{V}_{+,0,0}\right], \qquad P_{0,0} = \begin{pmatrix} 1 & 1 \\ 1 & 1 \end{pmatrix}. \tag{2.17}$$

Furthermore, we will group the contributions of $\epsilon$ and $\sigma$ using the relation $\lambda_{\sigma\sigma\epsilon} = \lambda_{\sigma\epsilon\sigma}$. We will work in $d = 3$ and assume that all other scalars apart from $\epsilon$ and $\sigma$ are irrelevant, so all remaining $\mathcal{O}^{\pm}$ will satisfy the spectrum restrictions:

$$S_+ = \{(\Delta, 0) : \Delta \geqslant 3\} \cup \{(\Delta, \ell) : \ell = 2, 4, 6, \ldots \text{ and } \Delta \geqslant \ell + 1\} \tag{2.18}$$

$$S_- = \{(\Delta, 0) : \Delta \geqslant 3\} \cup \{(\Delta, \ell) : \ell = 1, 2, 3, \ldots \text{ and } \Delta \geqslant \ell + 1\} \tag{2.19}$$

Then we can write (2.14) as

$$\vec{V}_{0,0} + \text{Tr}\left[P_{\Delta_\epsilon,0}\left(\vec{V}_{+,\Delta_\epsilon,0} + \begin{pmatrix} 1 & 0 \\ 0 & 0 \end{pmatrix}\vec{V}_{-,\Delta_\sigma,0}\right)\right]$$

$$+ \sum_{(\Delta,\ell)\in S_+} \text{Tr}\left[P_{\Delta,\ell}\vec{V}_{+,\Delta,\ell}\right] + \sum_{(\Delta,\ell)\in S_-} p_{\Delta,\ell}\vec{V}_{-,\Delta,\ell} = 0. \tag{2.20}$$

If the point $(\Delta_\sigma, \Delta_\epsilon)$ is allowed, this equation must have a solution with $P_{\Delta_\epsilon,0}, P_{\Delta,\ell} \succcurlyeq 0$, $p_{\Delta,\ell} \geqslant 0$. As discovered in [14],[11] this condition gives rise to an allowed region in the $(\Delta_\sigma, \Delta_\epsilon)$ plane consisting of a small island containing the 3d Ising CFT and a larger detached "continent." We will first discuss how this can be reproduced using a two-parameter navigator $\mathcal{N}(\Delta_\sigma, \Delta_\epsilon)$. See Section 2.2.1 below for how to include the third parameter $\theta$ parametrizing the ratio of the OPE coefficients $\lambda_{\sigma\sigma\epsilon}/\lambda_{\epsilon\epsilon\epsilon}$.

---

[11]Ref. [14] did not impose the constraint $\lambda_{\sigma\sigma\epsilon} = \lambda_{\sigma\epsilon\sigma}$ so their allowed region was somewhat larger than the one we will find. See [15], Eq. (2.3) for the setup we are describing here.

Analogously to (2.3), we consider the modification of (2.20) adding to the l.h.s. an extra term $\lambda \vec{M}$ where $\lambda \in \mathbb{R}$ and $\vec{M}$ is a particular 5-vector of functions of $u, v$:

$$\vec{V}_{0,0} + \lambda \vec{M} + \text{Tr}\left[ P_{\Delta_\epsilon,0} \left( \vec{V}_{+,\Delta_\epsilon,0} + \begin{pmatrix} 1 & 0 \\ 0 & 0 \end{pmatrix} \vec{V}_{-,\Delta_\sigma,0} \right) \right]$$
$$+ \sum_{(\Delta,\ell) \in S_+} \text{Tr}\left[ P_{\Delta,\ell} \vec{V}_{+,\Delta,\ell} \right] + \sum_{(\Delta,\ell) \in S_-} p_{\Delta,\ell} \vec{V}_{-,\Delta,\ell} = 0 \,. \quad (2.21)$$

In general $\vec{M}$ will also have some dependence on $\Delta_\sigma$ and $\Delta_\epsilon$ (just like all the other vectors in the equation). We will be looking for solutions of (2.21) with $P_{\Delta_\epsilon,0}, P_{\Delta,\ell} \succcurlyeq 0$ and $p_{\Delta,\ell} \geqslant 0$. Analogously to (2.4) and (2.5), the navigator $\mathcal{N}(\Delta_\sigma, \Delta_\epsilon)$ is defined as the minimal $\lambda$ such that a solution exists:

$$\mathcal{N}(\Delta_\sigma, \Delta_\epsilon) = \min \lambda \text{ such that (2.21) has a solution}, \quad (2.22)$$

while $\vec{M}$ has to be chosen such that there is always some solution for a sufficiently large $\lambda$. This then provides an upper bound for the navigator and in particular guarantees that $\mathcal{N} < +\infty$.

The GFF-navigator idea from Section 2.1.1 generalizes to the present multiple-correlator setup. Indeed, we always have a GFF solution to crossing in which $\sigma$ and $\epsilon$ are independent GFFs. The vector $\vec{M}$ is constructed from the contributions of (unit-normalized) operators $\frac{1}{\sqrt{2}} : \sigma^2 : \in \sigma \times \sigma$, $\frac{1}{\sqrt{2}} : \epsilon^2 : \in \epsilon \times \epsilon$, $: \sigma\epsilon : \in \sigma \times \epsilon$:

$$\vec{M}_{\text{GFF}} = \text{Tr}\left[ \begin{pmatrix} 2 & 0 \\ 0 & 0 \end{pmatrix} \vec{V}_{+,2\Delta_\sigma,0} \right] + \text{Tr}\left[ \begin{pmatrix} 0 & 0 \\ 0 & 2 \end{pmatrix} \vec{V}_{+,2\Delta_\epsilon,0} \right] + \vec{V}_{-,\Delta_\sigma+\Delta_\epsilon,0} \,. \quad (2.23)$$

With this $\vec{M}$, Eq. (2.21) has a solution with $\lambda = 1$, $P_{\Delta_\epsilon,0} = 0$ and $P_{\Delta,\ell}$ and $p_{\Delta,\ell}$ coming from the rest of the GFF spectrum in the $\sigma \times \sigma$, $\epsilon \times \epsilon$, $\sigma \times \epsilon$ OPE. This guarantees that $\mathcal{N}_{\text{GFF}}(\Delta_\sigma, \Delta_\epsilon) \leqslant 1$.[12]

To describe $\Sigma$-navigators we choose two finite sets $R_\pm \subset S_\pm$ of $(\Delta, \ell)$ pairs, and the linear equation

$$\vec{\mathbf{V}}_{0,0} + \sum_{(\Delta,\ell) \in R_+} \text{Tr}\left[ X_{\Delta,\ell} \vec{\mathbf{V}}_{+,\Delta,\ell} \right] + \sum_{(\Delta,\ell) \in R_-} x_{\Delta,\ell} \vec{\mathbf{V}}_{-,\Delta,\ell} = 0 \,, \quad (2.24)$$

where the variables $X_{\Delta,\ell}$ and $x_{\Delta,\ell}$ don't have to satisfy any positivity requirement. As in Section 2.1.2, the boldface symbols mean that we have switched to working at

---

[12]We used here the fact that all the GFF operators apart from $\sigma^2$, $\epsilon^2$, $\sigma\epsilon$ satisfy the $S_\pm$ constraints, assuming as we are that $\Delta_\sigma, \Delta_\epsilon \geqslant 1/2$. This is obvious for operators of spin $\ell \geqslant 1$ where we only impose the unitarity bounds. In the scalar sector, the next GFF operators are schematically $\sigma\Box\sigma$, $\epsilon\Box\epsilon$ and $\sigma\Box\epsilon$, all of which have dimension above 3. If there were additional GFF operators violating gap assumptions, their contributions would have to be added to (2.23). See App. A for an example. There it is also explained how to deal with the case where the navigator function depends on the magnitude of a squared OPE coefficient.

some finite order in Taylor expansion. Taking into account the structure of $\vec{V}_{0,0}$, $\vec{V}_{\pm,\Delta,\ell}$, $\vec{V}_{+,\Delta,\ell}$, and the fact that the functions $F^{ij,kl}_{\pm,\Delta,\ell}(u,v)$ are generically linearly independent (as follows from their expressions in [14]), Eq. (2.24) has a solution as long as $R_\pm$ include sufficiently many points.[13] We won't need to know anything about the solution apart from the fact that it exists.

So let us pick any two such sets $R_\pm$ with sufficiently many points, and define

$$\vec{M}_\Sigma = - \sum_{(\Delta,\ell)\in R_+} \mathrm{Tr}\left[C_{\Delta,\ell}\vec{V}_{+,\Delta,\ell}\right] - \sum_{(\Delta,\ell)\in R_-} c_{\Delta,\ell}\vec{V}_{-,\Delta,\ell}\,, \qquad (2.25)$$

with some strictly positive fixed coefficients $C_{\Delta,\ell} \succ 0$, $c_{\Delta,\ell} > 0$. For any such $\vec{M}_\Sigma$, Eq. (2.21) has a solution with some positive $\lambda$, by the same argument as in Section 2.1.2. Hence the corresponding $\Sigma$-navigator defined via (2.22) will be bounded from above.

As a final comment, we would like to recall another problem with the feasibility-mode searches which is resolved by our navigators. Feasibility-mode SDPB runs may not converge due to precision issues for points that can already be excluded using the bootstrap of crossing equations involving only a subset of the correlators [16]. E.g. this sometimes happens for points outside the 3d Ising island which are excluded by a single-correlator constraint. The navigators presented in this section converge in all the cases we tested, including the exact Ising setup that does exhibit this problem when run in feasibility-mode. Thus, navigators also provide a more robust method of checking the feasibility of any point.

### 2.2.1 Including the angles

As shown in [15], the allowed region in the 3-correlator bootstrap can be further reduced by treating the $P_{\Delta_\epsilon,0}$ term in (2.20) differently from the other $P_{\Delta,\ell}$. This is possible since we are assuming $\epsilon$ is non-degenerate. Writing $\lambda_{\sigma\sigma\epsilon} = \lambda_\epsilon \cos\theta$, $\lambda_{\epsilon\epsilon\epsilon} = \lambda_\epsilon \sin\theta$, $p_\epsilon = \lambda_\epsilon^2 \geqslant 0$, we can then specialize Eq. (2.20) as

$$\vec{V}_{0,0} + p_\epsilon \vec{V}_\epsilon(\theta) + \sum_{(\Delta,\ell)\in S_+} \mathrm{Tr}\left[P_{\Delta,\ell}\vec{V}_{+,\Delta,\ell}\right] + \sum_{(\Delta,\ell)\in S_-} p_{\Delta,\ell}\vec{V}_{-,\Delta,\ell} = 0\,, \qquad (2.26)$$

$$\vec{V}_\epsilon(\theta) = \mathrm{Tr}\left[\begin{pmatrix} c_\theta^2 & c_\theta s_\theta \\ c_\theta s_\theta^2 & s_\theta \end{pmatrix} \vec{V}_{+,\Delta_\epsilon,0} + \begin{pmatrix} c_\theta^2 & 0 \\ 0 & 0 \end{pmatrix} \vec{V}_{-,\Delta_\sigma,0}\right]\,. \qquad (2.27)$$

The original numerical implementation of this setup [15] involved scanning over the angle $\theta$ in addition to $\Delta_\sigma$ and $\Delta_\epsilon$, which was computationally laborious. Significant progress in reducing the computational cost has been recently achieved via the cutting surface algorithm [8].

In this paper we will show how this setup can be analyzed even more efficiently using the navigator function. The construction is almost the same as above. We simply add

---

[13]Generically it will suffice to take $|R_+| = \min(t_1, t_2, t_4 + t_5)$, $|R_-| = t_3$, where $t_i$ is the number of Taylor coefficients retained for line $i = 1\ldots 5$ of the original equation (2.14).

to the l.h.s. of (2.20) the term $\lambda \vec{M}$ and define the navigator $\mathcal{N}(\Delta_\sigma, \Delta_\epsilon, \theta)$ as the minimal value of $\lambda$ for which the so modified equation has a solution with $p_\epsilon \geqslant 0$, $P_{\Delta,\ell} \succcurlyeq 0$, $p_{\Delta,\ell} \geqslant 0$. We can choose $\vec{M}_{\text{GFF}}$ as in (2.23), or $\vec{M}_\Sigma$ as in (2.25), with $R_\pm \subset S_\pm$. The numerical results will be shown below.

### 2.2.2 Dual picture

The primal definition of the navigator function given above was convenient for clarifying the condition under which the navigator is bounded from above. For the actual numerical computation, we translate the primal definition to an equivalent dual formulation. As an example, for the 2-parameter navigator $\mathcal{N}(\Delta_\sigma, \Delta_\epsilon)$, Eq. (2.22), the dual definition takes the form:

$$\mathcal{N}(\Delta_\sigma, \Delta_\epsilon) = \max \ \vec{\alpha} \cdot \vec{V}_{0,0} \text{ over all linear functionals } \vec{\alpha} \text{ such that}$$

$$\vec{\alpha} \cdot \vec{M} = -1 \,, \tag{2.28}$$

$$\vec{\alpha} \cdot \left( \vec{V}_{+,\Delta_\epsilon,0} + \begin{pmatrix} 1 & 0 \\ 0 & 0 \end{pmatrix} \vec{V}_{-,\Delta_\sigma,0} \right) \succcurlyeq 0 \,, \tag{2.29}$$

$$\vec{\alpha} \cdot \vec{V}_{+,\Delta,\ell} \succcurlyeq 0 \text{ for all } (\Delta, \ell) \in S_+ \,, \tag{2.30}$$

$$\vec{\alpha} \cdot \vec{V}_{-,\Delta,\ell} \geqslant 0 \text{ for all } (\Delta, \ell) \in S_- \,. \tag{2.31}$$

For the 3-parameter navigator $\mathcal{N}(\Delta_\sigma, \Delta_\epsilon, \theta)$ from Section 2.2.1 we have to simply replace condition (2.29) with (see (2.27))

$$\vec{\alpha} \cdot \vec{V}_\epsilon(\theta) \geqslant 0 \,. \tag{2.32}$$

We recall that the above dual problems can be then transformed into a polynomial matrix problem using rational approximations of conformal blocks expanded up to some finite derivative order around the $z = \bar{z} = 1/2$ point. This polynomial matrix problem is then transformed into a semidefinite programming problem, which can be solved by SDPB [5, 6].

In App. B we describe an alternative construction of the navigator function, which turns the feasibility problem into an optimization problem not at the level of crossing equations, but after the problem has already been dualized and translated into an SDP. We have not used that construction in this work, but it may turn out useful in future applications.

## 3 Visualizing the GFF-navigator

In the previous section we provided a formal definition of navigator functions. Their actual numerical evaluation can be performed using SDPB. Since navigator evaluation involves maximization, it will be comparable in cost to an OPE coefficient maximization, and more expensive than say testing feasibility of a point. Of course, we hope that this

extra cost will be offset due to additional information provided by the navigator. And indeed, in subsequent sections we will see that complicated bootstrap tasks can be achieved with relatively few navigator evaluations.

Before we go to those applications, in this section we will explicitly visualize the various navigator functions of Section 2. We will do this to get some intuition about their "shape," and to check that they are sufficiently well behaved to allow application of minimization algorithms. Visualization will be done by performing fine scans in all variables. We emphasize again that in realistic applications we will not need to perform such expensive visualization scans.

We will focus on the 2- and 3-parameter GFF-navigators $\mathcal{N}(\Delta_\sigma, \Delta_\epsilon)$ and $\mathcal{N}(\Delta_\sigma, \Delta_\epsilon, \theta)$ from Sections 2.2 and 2.2.1. Numerical evaluation is done using the dual formulations given in Section 2.2.2, where we need to put $\vec{M} = \vec{M}_{\mathrm{GFF}}$ from Eq. (2.23). We will not show plots for the $\Sigma$-navigators, although we have checked that they behave similarly to the GFF-navigators.

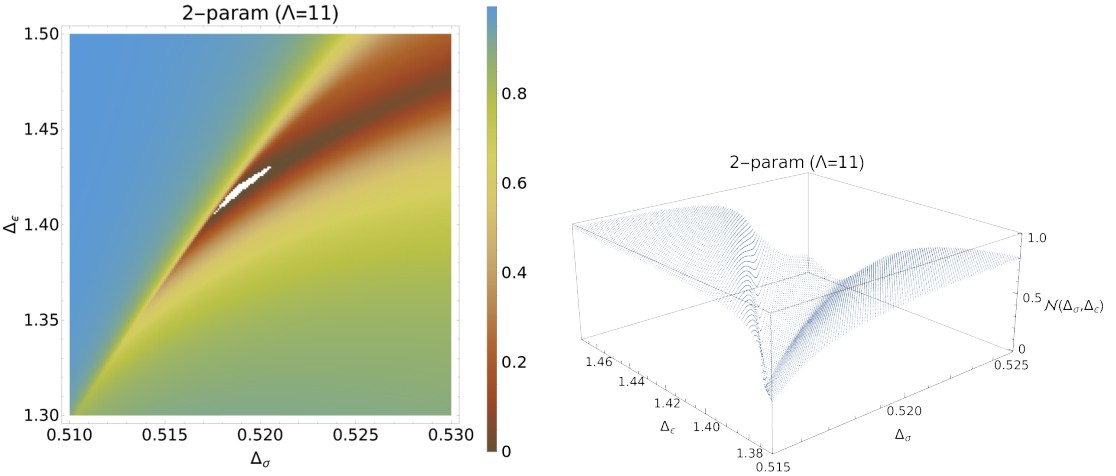

Figure 1: Example of a navigator function $\mathcal{N}(\Delta_\sigma, \Delta_\epsilon)$ for the 3d Ising setup. *Left:* Heat map of the navigator function. The negative region, corresponding to the Ising model island, is depicted in white. (Note that this image, and similarly other heat maps in this paper, appears pixelated due to the finite resolution of our scan. The actual island has a piecewise smooth boundary.) *Right:* Surface plot of the navigator function.

**2-parameter case.** We start with Fig. 1 showing $\mathcal{N}(\Delta_\sigma, \Delta_\epsilon)$ in an extended region around the 3d Ising island at the derivative order $\Lambda = 11$. We can see from it that the region of negative navigator value matches in size the $\Lambda = 11$ allowed region of [14], Figs. 3 and 4.[14] On this scale the navigator is observed to be smooth (see however below) and approaching its predicted asymptotic value $\mathcal{N}_{\mathrm{max}} = 1$ far away from allowed regions. There is clearly a valley coming from the top right of Fig. 1(left), narrowing to a tight gorge as it approaches its minimum inside the island. The surface has only

---

[14]Our $\Lambda = 11$ corresponds to $n_{\mathrm{max}} = 6$ in [14]. The slight difference in shape between our island and that of [14] is because we have imposed the OPE equality $\lambda_{\sigma\sigma\epsilon} = \lambda_{\sigma\epsilon\sigma}$ in our setup, see footnote 11.

one local minimum located in the plotted region and, as expected, it is inside the island. This feature will be essential when we discuss navigator minimization strategies in Section 5. Indeed, local minima in the disallowed region would have required more computationally expensive optimization methods than the BFGS algorithm discussed there.

In addition to the island, the allowed region found in [14] also included a detached "continent" at larger values of $\Delta_\sigma$, beyond the range of Fig. 1. This continent is of course also found to be a region of negative navigator. Our navigator minimization strategies will use a bounding box, see Section 5.2, to make sure that we sail to the island and not to the continent.

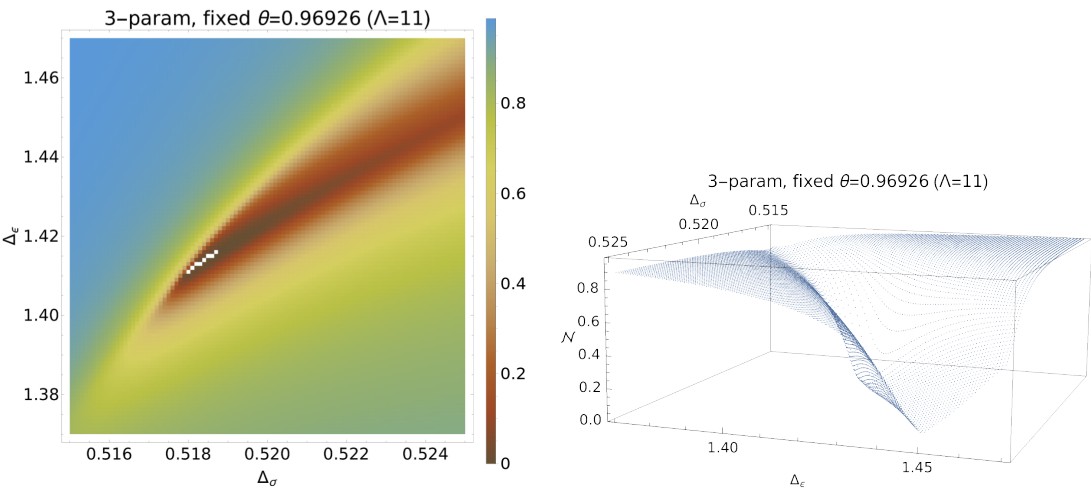

Figure 2: $(\Delta_\sigma, \Delta_\epsilon)$ slice of the 3-parameter GFF navigator $\mathcal{N}(\Delta_\sigma, \Delta_\epsilon, \theta = 0.96926)$ at $\Lambda = 11$. *Left:* Heat map of this 2d slice. *Right:* Surface plot of the 2d slice.

**3-parameter case.** To get an idea of the shape of $\mathcal{N}(\Delta_\sigma, \Delta_\epsilon, \theta)$, we will show two-dimensional slices for fixed values of one of the 3 parameters. Thus, in Fig. 2 we fix $\theta = 0.96926$ (the central value from [15]), and let $(\Delta_\sigma, \Delta_\epsilon)$ vary in a region close to the navigator minimum. The surface shape is similar to the two-parameter navigator surface in Fig. 1.[15]

Furthermore, in Fig. 3 we show 2d slices of the 3-parameter navigator arising for a fixed $\Delta_\sigma$ and $\Delta_\epsilon$. Although the precise shapes here are somewhat different, all three 2d slice surfaces are found to be smooth at this scale and free of local minima in the disallowed region (i.e. where the navigator is positive). This is a good sign that optimization algorithms should be able to quickly converge towards the Ising island given a reasonably precise initial guess.

**Variation with $\Lambda$.** Here we will explore how navigator shape changes with the derivative order $\Lambda$. By design, the navigator function monotonically increases pointwise with $\Lambda$, i.e. $\mathcal{N}_{\Lambda_2}(x) \geqslant \mathcal{N}_{\Lambda_1}(x)$ for $\Lambda_2 > \Lambda_1$. This generalizes the fact that the allowed region shrinks with $\Lambda$. It is interesting to know *how* this increase happens. E.g. does

---

[15]The surface plot in Fig. 2 is rotated opposite to Fig. 1, to facilitate comparison to Fig. 4 below.

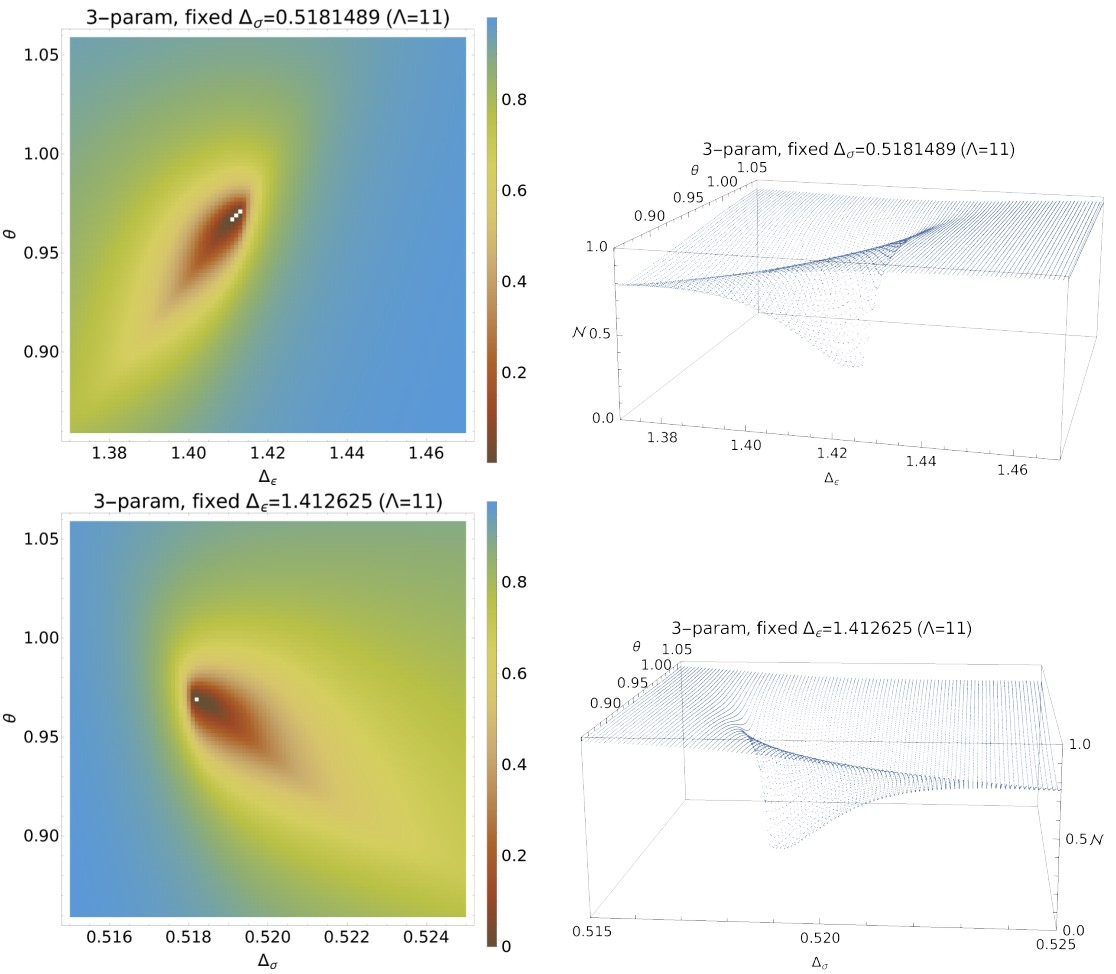

Figure 3: *Top row:* 2d slice of the 3-parameter GFF-navigator for fixed $\Delta_\sigma = 0.5181489$ around the Ising island at $\Lambda = 11$. *Bottom row:* Same, but for fixed $\Delta_\epsilon = 1.412625$.

the navigator surface move up with $\Lambda$ uniformly or not? To answer this question, we show in Fig. 4 the 2d slice of the 3-parameter navigator at fixed $\theta = 0.96926$ with $\Lambda = 19$, comparing it to $\Lambda = 11$ from Fig. 2. We see that the navigator surface has indeed moved up, but in non-uniform fashion. Most notably, the surface along one of the nearly flat "valley" directions gets lifted up much more than near the minimum. As a result, the minimum became more pronounced, which is a good sign.

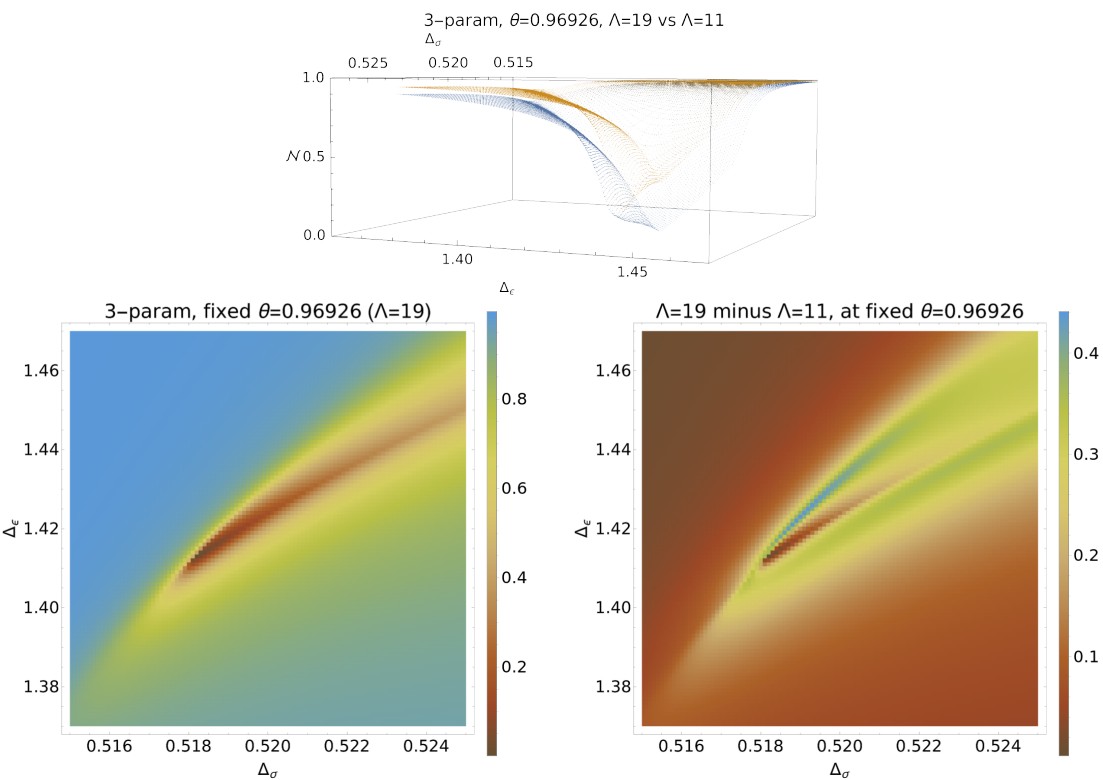

Figure 4: *Top:* Surface plot of the $\Lambda = 19$ 2d slice (orange) compared to the $\Lambda = 11$ 2d slice from Fig. 2 (blue) *Bottom left:* Heat map of the $(\Delta_\sigma, \Delta_\epsilon)$ slice of the 3-parameter GFF navigator $\mathcal{N}(\Delta_\sigma, \Delta_\epsilon, \theta = 0.96926)$ around the Ising island at $\Lambda = 19$. *Bottom right:* Heat map of the difference between $\Lambda = 19$ and $\Lambda = 11$.

## 3.1 Derivative of the navigator

The visualizations show navigator functions that are seemingly smooth and free of local minima. Both these properties would be very helpful for the numerical minimization algorithms, but they did not automatically follow from the definition of the navigator functions and we cannot guarantee that they hold in other setups. In fact, in the course of our investigations we found that even the navigator function under consideration is not *entirely* smooth: more precisely, we believe that it is not everywhere $C^2$.

Our evidence is provided in figure 5. In this figure we consider a GFF navigator function with $\Lambda = 11$ for $\Delta_\sigma = 0.51831848513294$, as a function of $\Delta_\epsilon$. (The chosen

values of $\Delta_\sigma$ and $\Delta_\epsilon$ are in the vicinity of the minimum that we found using the techniques described below. Notice that the navigator is negative along the entirety of the cross-section in figure 5 and so we are inside the Ising island. We also imposed the OPE relation $\lambda_{\sigma\sigma\epsilon} = \lambda_{\sigma\epsilon\sigma}$ but left the ratio $\lambda_{\epsilon\epsilon\epsilon}/\lambda_{\sigma\epsilon\sigma}$ unspecified.) We plot both the navigator function itself as well as its first derivative in the $\Delta_\epsilon$ direction. The kink in the latter plot strongly suggests that there is a discontinuity in the second derivative of the navigator. Indeed, the straight lines on either side of the kink allow us to reliably estimate the second derivative with finite differences: we find the value to be 767.762901557722(1) on the left and 219229.421457(1) on the right. Furthermore, using the two points closest to the kink we can estimate that the third derivative would have to be at least $10^{23}$ if the navigator function were smooth, which seems highly unlikely.

Although we have only shown a single cross section plot, it is likely that the non-smoothness persists along a line (segment) in the $(\Delta_\sigma, \Delta_\epsilon)$ plane. It would be interesting to understand its origin and whether there is a connection with the physics of the problem. Some preliminary investigations indicate that the discontinuity might be due to rearrangements of the extremal spectrum, but a detailed investigation is beyond the scope of this work.

Fortunately we will see below that the jump in the second derivative does not appear to inhibit the functioning of our minimization algorithm. We will comment more on this in the section 5.3.

# 4 Gradient at primal-dual optimality

In order to find points $x$ where $\mathcal{N}(x) < 0$ we will use a numerical minimization algorithm. The convergence rate of such algorithms is significantly improved if we also provide it with derivative information. In this section we therefore outline a procedure to compute the gradient $\nabla\mathcal{N}(x)$.

Naively, one might think that gradient evaluation would involve computational overhead. For example, evaluating it via finite differences would require $k$ additional SDPB runs where $k$ is the number of variables on which the navigator depends. However this naive expectation is wrong: the main result of this section will be that $\nabla\mathcal{N}(x)$ can be evaluated at negligible computational cost if we have already evaluated the function $\mathcal{N}(x)$ itself. The underlying reason is that the evaluation of $\mathcal{N}(x)$ is an extremization problem, and at extremality the first-order variation can be computed using only the original, unperturbed solution. This remains true even for constrained minimization problems, as is the case for us, when solved via primal-dual algorithms such as in SDPB, because primal and dual variables play the role of each other's Lagrange multipliers. To explain this in more detail we first have to introduce the semidefinite programming problem that underlies the computation of $\mathcal{N}(x)$.

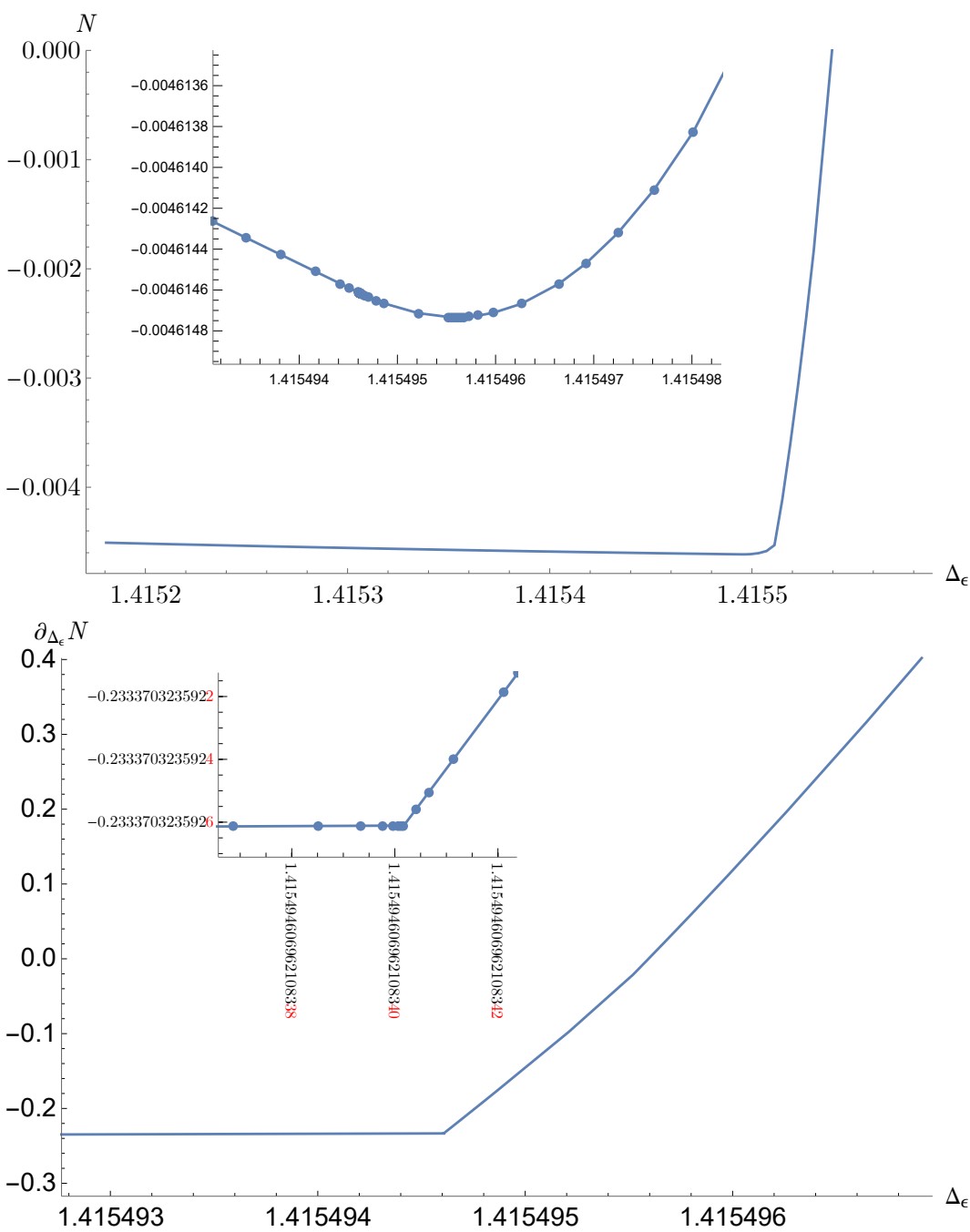

Figure 5: *Top:* Plot of $N(\Delta_\sigma, \Delta_\epsilon)$ v.s. $\Delta_\epsilon$ where $N$ is the navigator function, and a zoom-in plot around the minimum. *Bottom:* Plot of the derivative of the navigator function with respect to $\partial_{\Delta_\epsilon} N(\Delta_\sigma, \Delta_\epsilon)$ as a function of $\Delta_\epsilon$ and a zoom-in plot in the scale of $10^{-18}$ around the kink.

## 4.1 Semidefinite programming reminder

Now we will explain how to compute the gradient of the objective in the above setup. As mentioned above, the evaluation of $\mathcal{N}(x)$ is computationally analogous to an OPE extremization problem that is often encountered in numerical bootstrap studies. Let us recall that, using a rational approximation for conformal blocks [17], these extremization problems become semidefinite programs with a particular structure of the constraint matrices. We will use the notation of [5], using which the problem can be written as:

$$\mathcal{D} : \text{maximize } b^T y \quad \text{over} \quad y \in \mathbb{R}^n, Y \in \mathcal{S}^K$$
$$\text{such that } Y \succeq 0 \quad \text{and} \tag{4.1}$$
$$By + \text{Tr}(A_* Y) = c \,,$$

with $\mathcal{S}^K$ the space of symmetric matrices of size $K$. Note that $c \in \mathbb{R}^P$ is a vector, $B \in (\mathbb{R}^n)^P$ a rectangular matrix, and the $A_* = (A_1, \ldots, A_P) \in (S^K)^P$ is a vector of matrices.[16]

In the language of convex optimization the program (4.1) is called a *dual* program ($\mathcal{D}$), and the corresponding *primal* program $\mathcal{P}$ is given by:[17]

$$\mathcal{P} : \text{minimize } c^T x \quad \text{over} \quad x \in \mathbb{R}^P$$
$$\text{such that } X(x) := x^T A_* \succeq 0 \quad \text{and} \tag{4.2}$$
$$B^T x = b \,.$$

Note that $x^T A_* \equiv \sum_{p=1}^P x_p A_p$, so that $X(x) \in \mathcal{S}^K$.

We need a few more definitions. A vector $x$ is said to be *primal feasible* if all the conditions in (4.2) are obeyed, even if optimality is not necessarily achieved. In the same vein a pair $(y, Y)$ can be *dual feasible* if it obeys all the conditions in (4.1). The *duality gap* is defined as the difference between the objectives:

$$D(x, y) := c^T x - b^T y \,. \tag{4.3}$$

If $x$ is primal feasible and $(y, Y)$ is dual feasible, then the duality gap is nonnegative:

$$D(x, y) = \text{Tr}\big((x^T A_*) Y\big) = \text{Tr}(XY) \geqslant 0 \,, \tag{4.4}$$

by the positive semidefiniteness of $X$ and $Y$. So for any primal feasible point $x$ the value of $c^T x$ provides an upper bound for the dual optimum, and similarly for any dual feasible point $(y, Y)$ the value of $b^T y$ provides a lower bound for the primal optimum.

---

[16]Although this notation suffices for our purposes, in actuality the matrices involved all have a block structure and the number of non-zero components is significantly lower than a naive counting would suggest.

[17]We have opted to keep in this section the notation of [5] (excepting setting $C = 0$ in Eq. (2.3) and (2.21) of [5]). This unfortunately produces a clash of notation: in this section $x$ denotes the vector of free variables in the primal semidefinite program, whereas in the rest of the paper $x$ is the argument of the navigator function. We stress that these are unrelated quantities.

Now suppose one finds primal and dual feasible points with $D(x, y) = 0$. Then clearly both the primal and dual problem have been solved and brought to extremality, because neither objective has any room left to improve. It is a non-trivial fact of life that this condition is not only sufficient but also necessary for optimality in a generic semidefinite program (see [5] and references therein for details). In other words, rather than solving the primal or dual extremization problem, we can equivalently solve

$$\begin{aligned} \text{Tr}\,(A_* Y) + By &= c\,,\\ B^T x &= b\,,\\ X &= x^T A_*\,,\\ XY &= 0\,,\\ X, Y &\succeq 0\,, \end{aligned} \tag{4.5}$$

and then the optimal value of (4.1) and (4.2) is given by $b^T y = c^T x$. Notice that the fourth equation in (4.5) states that $XY = 0$ as a matrix equation. We call this the *complementarity condition*, and it follows from the vanishing duality gap, i.e. $\text{Tr}(XY) = 0$, together with $X, Y \succeq 0$.

## 4.2 SDP gradient formula

Suppose we have found a primal-dual optimal point $(x, y, X, Y)$ such that the equations (4.5) are solved. To compute the gradient of the objective we change the parameters in the problem a little bit,

$$(b, c, B, A_*) \to (b, c, B, A_*) + (db, dc, dB, dA_*)\,, \tag{4.6}$$

and ask how the objective will change. So we need to investigate the corresponding linearized problem. The change in the solution

$$(x, y, X, Y) \to (x, y, X, Y) + (dx, dy, dX, dY) \tag{4.7}$$

must obey the linearized version of the optimality equations (4.5):

$$\begin{aligned} \text{Tr}(dA_* Y) + \text{Tr}(A_* \, dY) + B\, dy + dB\, y &= dc\,,\\ dB^T x + B^T dx &= db\,,\\ dX &= dx^T A_* + x^T dA_*\,,\\ dX\, Y + X\, dY &= 0\,,\\ X + dX,\ Y + dY &\succeq 0\,. \end{aligned} \tag{4.8}$$

Our goal will be to compute the change in the dual objective, which is given by:

$$d(b^T y) = db^T y + b^T dy\,. \tag{4.9}$$

In fact, since the duality gap remains zero we find $d(c^T x) = d(b^T y)$ and one could equally well have computed the change in the primal objective.

We start by showing a useful auxiliary result. The $dX Y + X dY = 0$ in (4.8) implies of course that $\text{Tr}(dX Y) + \text{Tr}(X dY) = 0$. We claim that a stronger result is true, namely that the two terms vanish independently:

$$\text{Tr}(dX Y) = \text{Tr}(X dY) = 0. \tag{4.10}$$

The proof is as follows. If $XY = 0$ and $X, Y \succeq 0$ then $X$ and $Y$ must have some zero eigenvalues. We can choose a basis where $X$ is an upper block matrix,

$$X = \begin{pmatrix} X_{11} & 0 \\ 0 & 0 \end{pmatrix} \tag{4.11}$$

with $X_{11} \succ 0$. Then any symmetric $Y$ obeying $XY = 0$ must look like

$$Y = \begin{pmatrix} 0 & 0 \\ 0 & Y_{22} \end{pmatrix} \tag{4.12}$$

with $Y_{22} \succeq 0$ because $Y \succeq 0$. If we now write the variations as

$$dX = \begin{pmatrix} dX_{11} & dX_{12} \\ dX_{12}^T & dX_{22} \end{pmatrix}, \qquad dY = \begin{pmatrix} dY_{11} & dY_{12} \\ dY_{12}^T & dY_{22} \end{pmatrix}, \tag{4.13}$$

then

$$dX Y = \begin{pmatrix} 0 & dX_{12}Y_{22} \\ 0 & dX_{22}Y_{22} \end{pmatrix}, \qquad X dY = \begin{pmatrix} X_{11}dY_{11} & X_{11}dY_{12} \\ 0 & 0 \end{pmatrix}. \tag{4.14}$$

Now it becomes clear that the condition $dX Y + X dY = 0$ implies that $X_{11}dY_{11} = 0$ and $dX_{22}Y_{22} = 0$, which in turn implies (4.10).

Let us return to the change in the dual objective as given in equation (4.9). Using the linearized optimality equations it can be written as:

$$\begin{aligned} db^T y + b^T dy &= db^T y + x^T B dy \\ &= db^T y + x^T (dc - \text{Tr}(dA_* Y) - \text{Tr}(A_* dY) - dB y) \\ &= db^T y + x^T dc - x^T \text{Tr}(dA_* Y) - x^T \text{Tr}(A_* dY) - x^T dB y \end{aligned} \tag{4.15}$$

At this point we recall that $x^T A_* = X$. Moreover we have just shown $\text{Tr}(X dY) = 0$. So the term proportional to $dY$ in (4.15) vanishes, and we obtain:

$$\boxed{d(b^T y) = d(c^T x) = db^T y + dc^T x - x^T dB y - x^T \text{Tr}(dA_* Y).} \tag{4.16}$$

This "SDP gradient formula" constitutes one of the main points of our paper. It shows that the variation of the objective function of semidefinite programs (4.1) and (4.2) can be computed just from the variation of the data $(db, dc, dB, dA_*)$ provided that we know the primal-dual solution $(x, y, X, Y)$. A remarkable fact is that we have eliminated all the dependence on $(dx, dy, dX, dY)$ from this formula.

In this work we will apply Eq. (4.16) to the navigator function. Once the navigator has been evaluated for some parameter values, Eq. (4.16) computes the gradient at the same point with negligible extra computational cost (see Section 4.2.1 below for how we organized the computation in practice). It's worth pointing out that this observation holds also for more familiar conformal bootstrap problems such as the OPE coefficient maximization. Such problems have been analyzed for years using primal-dual methods, but the existence of the "SDP gradient formula" has never been suspected by the people in the bootstrap community.

There is one important caveat to the preceding derivation. Although the solution $(dx, dy, dX, dY)$ to the linearized optimality equations does not appear in equation (4.16), we did need to assume that it existed in the intermediate steps. On the other hand, it is not guaranteed that the equations in (4.8) always have a solution. Fortunately this question has been analyzed in the semidefinite programming literature: for example, the paper [18] proves that the linearized equations for the semidefinite programs considered here will have a solution if $X + Y \succ 0$, which in our notation is equivalent to the requirement that $Y_{22} \succ 0$ rather than just $Y_{22} \succeq 0$. The paper [19] shows that this is in fact a generic property of the optimal matrices in semidefinite programs. We therefore expect the navigator functions to be generically $C^1$, which is also confirmed experimentally by the smooth plots shown in the previous section.

### 4.2.1 Practical details for navigator gradient evaluation

As was shown in the previous section the change of the objective under a small perturbation of a bootstrap problem can be computed using only the solution to the initial problem $(x, y, X, Y)$ and the differences between the data $(db, dc, dB, dA_*)$ defining the SDP. In this short technical section we describe the precise workflow using available codes. In order to be able to compute the gradient, one should first run SDPB on the original problem specified by $(b, c, B, A_*)$ using the option `--writeSolution="x,y,X,Y"` to save the full solution to a file. The values $(db, dc, dB, dA_*)$ can either be obtained by taking the difference between the perturbed and unperturbed bootstrap problem on the level of the polynomial matrix problem (PMP) and converting that to an SDP using `pvm2sdp` or by first converting both PMP's to SDP's and taking the difference between the resulting $(b, c, B, A)$ and $(b', c', B', A')$. For the computations in this paper we did the latter. A dedicated tool called `approx_objective` that takes one optimal checkpoint containing $(x, y, X, Y)$, one unperturbed SDP-file and one perturbed SDP-file[18] as input and outputs the corresponding change in the objective has been packaged with SDPB as of version 2.5.

When converting the PMP to an SDP, we can choose to keep the bilinear basis, sample points, and sample scalings (see [5]) the same for both the perturbed and the unperturbed SDP. With such choices, we automatically have $dA_* = 0$ and the gradient formula simplifies to $d(b^T y) = db^T y + dc^T x - x^T dB\, y$.

---

[18]The file is expected to contain $(b, c, B, A_*)$ in the format produced by `pvm2sdp`.

## 4.3 Lagrangian perspective

In this section we give an alternative derivation of the SDP gradient formula. This derivation may look like a trick, but it provides an interesting perspective on why we were able to eliminate the variation $(dx, dy, dX, dY)$ from the change $d(b^T y)$ in the dual objective.

Consider the following Lagrange function:

$$L(x, y, X, Y) = c^T x + b^T y - x^T B y + \text{Tr}\big((X - x^T A_*)Y\big) - \mu \log \det X \qquad (4.17)$$

with $\mu > 0$ a parameter, and it is understood that $X \succeq 0$. As is readily verified, the stationarity equations of this Lagrangian with respect to $x$, $y$ and $Y$ yield exactly the primal and dual feasibility conditions, i.e. the first three conditions in (4.5). Demanding stationarity with respect to $X$ yields:

$$XY = \mu I, \qquad (4.18)$$

with $I$ the identity matrix. We can think of the last term $-\mu \log \det X$ in (4.17) as a barrier function that guarantees that $X, Y \succ 0$. In the limit $\mu \downarrow 0$ the barrier disappears and we recover the original complementarity condition.[19] As is well known (e.g. [20]), the barrier function $-\log \det X$ is convex.

We denote by $(x(\mu), y(\mu), X(\mu), Y(\mu))$ the stationary point of the Lagrange function, i.e. the solution of all the feasibility conditions and of the deformed complementarity condition $XY = \mu I$. Apart from degenerate situations, this solution exists; it is also unique.[20] The value of the Lagrange function at this solution is given by:

$$L(\mu) = c^T x(\mu) - \mu \log \det X(\mu) \qquad (4.19)$$

since the constraints multiplying $y$ and $Y$ are obeyed by assumption. Furthermore, since the Lagrange function is stationary with respect to $(x, y, X, Y)$, its variation with respect to the parameters $(b, c, B, A_*)$ is immediate:

$$dL(\mu) = dc^T x(\mu) + db^T y(\mu) - x(\mu)^T \, dB \, y(\mu) - x(\mu)^T \, \text{Tr}(dA_* \, Y(\mu)) \qquad (4.20)$$

---

[19]The modified complementarity condition (4.18) is also at the heart of primal-dual interior point algorithms as used in SDPB. Keeping $\mu$ finite and therefore $X, Y$ strictly positive is useful to avoid getting stuck at the boundary where $X$ and $Y$ are singular. In the course of the algorithm the value of $\mu$ is then gradually reduced to zero in order to obtain a solution that obeys the original complementarity condition. See [5] for details.

[20]Let $M$ be the convex set of all $x$ obeying $b^T = x^T B$ and $x^T A_* = X(x) \succ 0$. On this set we consider the convex and smooth function $t(x) = c^T x - \mu \log \det X$. Generically the sublevel sets of this function are bounded. Indeed, any unbounded direction inside $M$ can be parametrized as $x_0 + \lambda \hat{x}$ with $x_0 \in M$, and $\hat{x}^T B = 0$ and with $\lambda \to \infty$. If the original primal minimization problem is bounded, we have $c^T \hat{x} \geqslant 0$ for all directions. Generically we have a stronger condition $c^T \hat{x} > 0$, in which case we eventually exit all sublevel sets for any such direction. Therefore $t(x)$ must have a minimum inside $M$, and by convexity it is unique. At this point $\nabla t(x) = c - \mu \text{Tr}(A_* X^{-1})$ is orthogonal to $M$. But the directions orthogonal to $M$ are spanned by the gradient of the constraints, so by the columns of the matrix $B$. There must then exists some coefficients $y$ such that $\nabla t(x) = By$, which solve the last remaining equation.

Now we can ask what happens if we take $\mu$ very small. Since the original semidefinite program is assumed to have a solution, we expect $(x(\mu), y(\mu), X(\mu), Y(\mu))$ to smoothly approach that solution as we send $\mu \downarrow 0$. Clearly, the variation of the Lagrangian (4.20) at the stationary point will then approach the right-hand side of our previous result (4.16). What of the value of the Lagrangian (4.19) itself? We know that $X$ becomes singular and so $\det(X)$ will diverge. However, for $Y = \mu X^{-1}$ to remain finite the eigenvalues of $X$ cannot vanish faster than linearly with $\mu$. We conclude that $-\mu \log \det X = O(\mu \log \mu)$, the additional term in equation (4.19) vanishes in the limit, and so $\lim_{\mu \downarrow 0} L(\mu) = c^T x$. Together with equation (4.20) this reproduces (4.16).

This derivation elucidates the absence of $(dx, dy, dX, dY)$ from the variation of the objective. To summarize, the point is to replace the original constrained problem with an unconstrained one, involving a barrier function times a regulator $\mu$. The unconstrained variation involves only variation of the data, and not of the solution itself. The constrained variation is recovered in the $\mu \downarrow 0$ limit and also has this property.

In Appendix C we push this logic one step further and explain how it can be used to compute the second variation of the objective, which one may call the "SDP Hessian formula." Also there we provide numerical tests of the SDP gradient and Hessian formulas. Having the Hessian as opposed to just the gradient could further speed up the minimization algorithms to be described in the next section, allowing to use Newton rather than quasi-Newton methods, but exploring this is postponed to future work.

# 5  Navigator minimization

A central task in the numerical bootstrap is the search for a feasible point. This corresponds to finding a point where the navigator function is negative. In addition, we may also be interested in finding the minimum of the navigator function, since its location might be close to the true CFT (we will show shortly that this indeed seems to be the case).

Given an $n$-dimensional search space, the search for a local minimum of the navigator function $\mathcal{N}(x)$ is a standard optimization problem. As explained in Section 4, the gradient of the navigator function is cheap to compute. Quasi-Newton methods can make good use of this cheap gradient. Recall that Newton's method requires computing a gradient and a Hessian at each point in the search. By contrast, quasi-Newton methods approximate the Hessian using gradient information.[21] In this work, we use the Broyden-Fletcher-Goldfarb-Shanno (BFGS) algorithm ([21], Sec. 6.1) which is a well documented and widely used quasi-Newton method.

The BFGS algorithm maintains an approximation to the Hessian, which it updates using gradient information at each step. This update enforces positive-definiteness of the Hessian. Thus, it can only provide a truthful representation of the Hessian if the objective function is convex. In the examples studied in this paper, we have found that

---

[21]In Appendix C we explain that it also possible to compute the Hessian of the navigator. However in this work we will only use the gradient information.

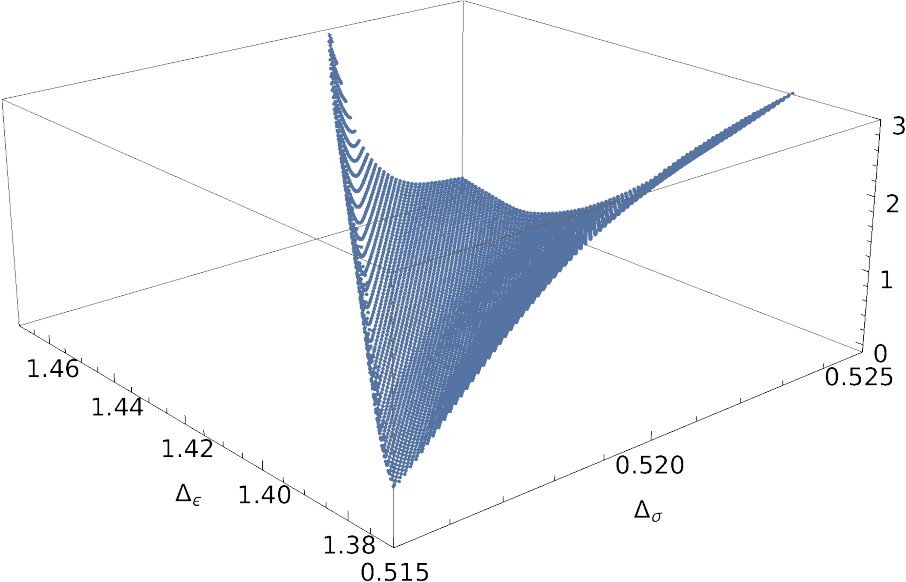

Figure 6: Rendering of $f\left(\mathcal{N}(\Delta_\sigma, \Delta_\epsilon)\right)$, i.e. the 2-parameter GFF navigator to which was applied the fractional linear transformation (5.1). The derivative order used here is $\Lambda = 11$.

the navigator function is convex close to its minimum. However, this is not true further away from the minimum (for example, the GFF navigator tends to its asymptotic value 1 in a concave manner far away from allowed regions). This can lead to failure of the BFGS algorithm or less than optimal convergence. Therefore it is helpful to compose the navigator function with a monotonic function so that it becomes convex in a larger region but maintains the same minima. For example, if the maximal value of the navigator $\mathcal{N}(x)$ is $\mathcal{N}_{\max}$ (e.g. $\mathcal{N}_{\max} = 1$ by construction for the GFF navigator), we can instead minimize

$$f(x) = \frac{\mathcal{N}(x)}{1 - \mathcal{N}(x)/\mathcal{N}_{\max}} \ . \tag{5.1}$$

Note that $f(x) < 0$ if and only if $\mathcal{N}(x) < 0$, so that the allowed region is unchanged after this transformation. It's also easy to show that $f(x)$ is convex in a larger region than $\mathcal{N}(x)$.[22] Intuitively, the main idea is that $f(x) \approx \mathcal{N}(x)$ where $\mathcal{N}(x) \approx 0$, while at large $x$, where $\mathcal{N}(x)$ approaches its asymptotic limit and hence is not convex, $f(x)$ instead grows and has a chance to be convex. E.g. if $\mathcal{N}_{\max} - \mathcal{N}(x) = O(|x|^{-a})$ at large $x$, then $f(x)$ grows as $|x|^a$, which is convex for $a > 1$.[23]

To see how this works in practice, consider the GFF-navigator plotted in Fig. 1, which is clearly not convex. Fig. 6 shows the result of applying to it transformation (5.1) with $\mathcal{N}_{\max} = 1$. We can see that the fractional linear transformation indeed improves the convexity of the objective function fed to BFGS. The function in Fig. 6 is

---

[22]For example, in the 1D case, we have $f''(x) \geqslant 0$ iff $\mathcal{N}''(x)[\mathcal{N}_{\max} - \mathcal{N}(x)] + [\mathcal{N}'(x)]^2 \geqslant 0$.

[23]If it turns out e.g. that $\mathcal{N}(x)$ approaches its asymptotic limit as an inverse power of $x$, but with $a < 1$, one could consider the modified function $f(x) = \frac{\mathcal{N}(x)}{(1-\mathcal{N}(x)/\mathcal{N}_{\max})^k}$ with $k > 1$.

still not globally convex, but it is locally convex, or close to it, in a much larger region than the original function in Fig. 1. We will see below that this transformation indeed results in more appropriate step lengths in the initial line searches and that BFGS has a higher rate of success of finding the Ising model minimum, even when starting in regions of relative flatness of the untransformed navigator.

In our studies we will use the standard implementation of the BFGS algorithm which can be found in the SciPy library [22], with some minor modifications. In Section 5.1, we review the BFGS method. We describe our modifications and their motivation in Section 5.2. We will see in Section 5.3 that the resulting algorithm gives good results when applied to the 3d Ising model case. Finally, we will comment in Section 5.4 on further possible improvements on our modified BFGS algorithm.

## 5.1   BFGS algorithm

Let $f(x)$ be the objective function to be minimized. The BFGS algorithm attempts to minimize $f(x)$ by taking successive steps $x_0 \to x_1 \to \ldots \to x_k \to \ldots$ , where step $k$ is taken using the information from an approximated quadratic model of the function at $x_k$. This approximate quadratic model is

$$f(x_k + \Delta x) \approx f(x_k) + \nabla f(x_k)^{\mathrm{T}} \Delta x + \frac{1}{2} \Delta x^{\mathrm{T}} B_k \, \Delta x \ , \qquad (5.2)$$

where $B_k$ is an approximation to the Hessian at $x_k$. After BFGS takes the $k^{\mathrm{th}}$ step $x_k \to x_{k+1}$, it determines the approximate Hessian at $x_{k+1}$ by updating the one at $x_k$ using only gradient information at $x_k$ and $x_{k+1}$. For the full updating formula, see (6.19) of [21], Sec. 6. The minimum of the quadratic model (5.2) is the so-called "Newton step"

$$p_k = -B_k^{-1} \nabla f(x_k) \ . \qquad (5.3)$$

In Newton's method, at each iteration the Newton step would be taken, so that $x_{k+1} = x_k + p_k$. In BFGS, the Newton step is replaced by a line search in the direction of $p_k$. An *exact* line search would correspond to

$$x_{k+1} = \arg \min_{\alpha > 0} \phi(\alpha), \qquad \phi(\alpha) := f(x_k + \alpha p_k) \, . \qquad (5.4)$$

In practice, one uses an *inexact* line search, which means that one looks for an "approximate" minimum of $\phi(\alpha)$ at $\alpha > 0$. It turns out that a rather rough approximation is sufficient for good performance of the algorithm. A typical termination criterion is the "strong Wolfe conditions:"

$$\phi(\alpha) \leqslant \phi(0) + \mu \, \alpha \, \phi'(0) \, , \qquad (5.5)$$
$$|\phi'(\alpha)| \leqslant \eta |\phi'(0)| \, . \qquad (5.6)$$

The first condition enforces that the function decreases sufficiently. The parameter $\mu$ controlling this is usually chosen to be very small. We used the default value $\mu = 10^{-4}$

implemented in SciPy. The second condition demands that the gradient decreases sufficiently. This is usually called the curvature condition, and it guarantees that the BFGS update to the Hessian maintains positive-definiteness,[24] which in turn implies that $p_{k+1}$ will be a decrease direction, allowing the algorithm to proceed. The parameter $\eta$ controlling the demanded decrease is usually chosen somewhat below 1. We used the default value $\eta = 0.9$, with satisfactory results. The SciPy BFGS algorithm used in this paper relies on the Moré-Thuente line search algorithm [24], a standard and robust algorithm for finding points obeying the strong Wolfe conditions.

Once an "accepted" point, i.e. a point obeying these conditions, is found, the Hessian is updated,[25] and the BFGS algorithm proceeds with its next step. The algorithm terminates once the norm of the gradient gets smaller than some value $g_{\mathrm{tol}}$ supplied by the user (we used $g_{\mathrm{tol}} = 10^{-5}$).

It's worth pointing out that in the line searches, the Newton step $\alpha = 1$ is used as the initial guess. Once the Hessian has been well-approximated (as may happen towards the end of the minimization run), the first step $\alpha = 1$ will usually be accepted, and a convergence similar to that of Newton's method is expected. On the contrary, $\alpha = 1$ may not be a good guess at the beginning of the run unless we have an idea of the typical size of the region in which the minimum is expected to lie. This is provided via a bounding box in our modified BFGS algorithm described below.

## 5.2 Modified BFGS algorithm

The BFGS algorithm requires an initial guess for the Hessian at the first step $B_0$. This guess is usually taken to be the identity, which does not take into account the different scalings of the different variables. This is often okay because the BFGS algorithm recovers scale information after a sufficient number of steps have been taken, i.e. once the Hessian approximation becomes accurate in all directions. However, we still found that if some idea of the scale of the problem is known, e.g. if we have a vague idea about the location of the allowed region, it is best to incorporate this information into the initial Hessian. By setting a well-scaled initial Hessian, an appropriate step length in the initial line searches can be achieved. (Recall that the initial line search step is always $\alpha = 1$ in the direction of $p_k$, and the length as well as the direction of this search clearly depends on $B_k$ via (5.3).) This will ensure that the BFGS algorithm explores the vicinity of the starting point rather than a much larger space—a crucial feature in cases where we are interested in one specific nearby local minimum. For example, when we want to study the 3d Ising model, we are not interested in studying the navigator in

---

[24]This condition in particular trivially implies $(x_{k+1} - x_k) \cdot (\nabla f(x_{k+1}) - \nabla f(x_k)) > 0$. The latter condition guarantees that the BFGS Hessian update preserve positive definiteness; one should be able to convince oneself that this is the case by inspection of (6.17) in [21]. See [23], Theorem 3.2.2 and the top of p.56 for a proof and discussion.

[25]Note that the line search will also involve evaluating the function and its gradient at several intermediate points along the direction $p_k$, until a point satisfying the strong Wolfe conditions is found. In the BFGS algorithm, information from those intermediate points is not used in any way to improve the Hessian.

the big allowed "continent" found at large $\Delta_\sigma$ [14]. We found that the BFGS algorithm may end up in this much larger feasible region unless an appropriately scaled initial Hessian is supplied.

One trick to set an appropriately scaled initial Hessian (based on [21], p.142) is the following: Compute the gradient at the initial point, and set $B_0$ to

$$B_0 = \|\nabla f(x_0)\| \operatorname{diag}\left(\frac{1}{\alpha_0^1}, \cdots, \frac{1}{\alpha_0^n}\right). \tag{5.7}$$

Then, from (5.3), the initial Newton step $\alpha = 1$ will result in probing the function at $x_0 - \operatorname{diag}(\alpha_0^1, \cdots, \alpha_0^n) \cdot \frac{\nabla f(x_0)}{\|\nabla f(x_0)\|}$. Hence the parameters $\alpha_0^i$ have the meaning of the characteristic desired $|\Delta x^i|$ during the initial step of the first line search. Alternatively, one could use the procedure described in Appendix C to explicitly compute the initial Hessian. However we do not advise this, since the Hessian for a point far away from the minimum could very well not provide an accurate scale for the problem, nor is the Hessian far away from the minimum likely to provide a more accurate starting point for the approximation of the Hessian at the minimum than an appropriately scaled diagonal matrix.[26]

Apart from specifying the initial Hessian, some minor modifications have to be made in order to apply the BFGS algorithm to conformal bootstrap problems. Firstly, the navigator function is naturally defined only in certain regions and not globally. Consider for example the case of the 3d Ising model. The navigator function is naturally defined only for $\Delta_\sigma$ and $\Delta_\epsilon$ above the unitarity bound. Similarly, when demanding the existence of exactly one relevant parity odd and even singlet, we restrict the domain of $\mathcal{N}(\Delta_\sigma, \Delta_\epsilon)$ to values $\Delta_\sigma$ or $\Delta_\epsilon$ below 3. Additionally, as discussed above, one might only be interested in minima or negative values that are located in a certain region around the starting point.

Hence, we ask the user to provide a bounding box for the search space, past which we do not allow the search to move. This constraint is implemented by altering the line search such that if a step outside of the bounding box would be taken, the maximal step in the same direction within the boundaries is taken instead. If this point on the edge is accepted, i.e. obeys the strong Wolfe conditions, we check whether the new search direction points inside or outside of the bounding box. If the new search direction points outside of the bounding box, but the gradient descent direction lies inside, the search direction is taken to be the gradient descent direction for the next step. If neither the initial search direction nor the negative gradient lie inside the bounding box, the search is terminated. The user should then either try a different initial point or change the bounding box.[27]

---

[26]On the contrary, having access to the exact rather than BFGS-approximated Hessian is expected to speed up the last stage of the minimization run, although we have not took advantage of this possibility in this work.

[27]Note that if this happens, it may mean that that boundary includes some part of the attraction basin for a minimum that lies outside the bounding box. In this case an alteration of the relevant boundary is probably advised.

**Input:** A navigator function $\mathcal{N}(x)$, an initial guess $x_0$, the bounding box coordinates $b_{\min}^i, b_{\max}^i$ and a value $g_{\text{tol}}$.

**Output:** The final point $x_f$ and the termination message.

**begin**

$\quad$ $f(x) = \frac{\mathcal{N}(x)}{1 - \mathcal{N}(x)/\mathcal{N}_{\max}}$

$\quad$ $\alpha_0^i = 0.2 \times (b_{\max}^i - b_{\min}^i)$

$\quad$ $B_0 = \|\nabla f(x_0)\| \operatorname{diag}(\frac{1}{\alpha_0^x}, \frac{1}{\alpha_0^y}, \cdots)$

$\quad$ $p_0 = -B_0^{-1} \nabla f(x_0)$

$\quad$ **while** $\|\nabla f(x_k)\| > g_{\text{tol}}$ **do**

$\qquad$ $\alpha = \operatorname{linesearch}(f, x_k, p_k, B_k)$

$\qquad$ $x_{k+1} = x_k + \alpha p_k$

$\qquad$ The hessian $B_k$ is updated to $B_{k+1}$

$\qquad$ The search direction $p_k$ is updated to $p_{k+1}$

$\qquad$ **if** $x_{k+1}$ *is at the boundary* **then**

$\qquad\qquad$ **if** $p_{k+1}$ *points back inside the bounding box* **then**

$\qquad\qquad\qquad$ continue

$\qquad\qquad$ **else**

$\qquad\qquad\qquad$ **if** $-\nabla f(x_{k+1})$ *points back inside the bounding box* **then**

$\qquad\qquad\qquad\qquad$ $p_{k+1} = -\nabla f(x_{k+1})$

$\qquad\qquad\qquad$ **else**

$\qquad\qquad\qquad\qquad$ **return** $x_{k+1}$ and the termination message *"Out of the bounding box"*

$\qquad\qquad\qquad$ **end**

$\qquad\qquad$ **end**

$\qquad$ **end**

$\qquad$ Optional: **if** $f(x_{k+1}) < 0$ **then**

$\qquad\qquad$ **return** $x_{k+1}$ and the termination message *"Found a negative point"*

$\qquad$ **end**

$\quad$ **end**

$\quad$ **return** $x_k$ and the termination message *"Minimum found: gradient is smaller than the tolerance"*

**end**

Algorithm 1: Modified BFGS algorithm

The provided bounding box is also used to specify the desired step lengths in the initial Hessian of (5.7). We found satisfactory results by setting the desired step lengths in each direction to be 20% of the supplied bounding box.

It is fair to ask how the user will know which bounding box to specify. We assume that the user has some idea of the range of parameters they want to explore. Results obtained at lower derivative order $\Lambda$ can also be used for guidance, as well as estimates of CFT data coming from other methods such as RG or Monte Carlo simulations.

The BFGS algorithm including these modifications is summarized as Algorithm 1.

## 5.3 Minimization results

To illustrate the effectiveness of our minimization algorithm, we apply it to the classic conformal bootstrap problem of finding an allowed point corresponding to the 3d Ising model using the system of correlators $\{\langle\sigma\sigma\sigma\sigma\rangle, \langle\sigma\sigma\epsilon\epsilon\rangle, \langle\epsilon\epsilon\epsilon\epsilon\rangle\}$, which contains the lowest dimensional $\mathbb{Z}_2$-odd scalar $\sigma$ and the lowest dimensional $\mathbb{Z}_2$-even scalar $\epsilon$, under the assumption that those operators are the only relevant ones, as described in Section 2.2. We will see that the navigator function enables us to locate an allowed point with a relatively small number of SDPB calls. Finding an allowed point naively by checking feasibility for a dense grid of points covering the search space would take orders of magnitude more SDPB calls.

Of course, in a decade of feasibility searches many useful tricks have been found to speed them up.[28] Still, we foresee that navigator-function methods will offer even better performance. They should eventually allow computations in more complicated setups involving an even higher number of parameters to scan over, such as e.g. bootstrapping the full system of $\sigma, \epsilon, \epsilon'$ 4pt functions, which were not possible to treat so far via feasibility-based methods.

### 5.3.1 2-parameter searches

We start with the 2-parameter case which is easier to visualize. So we minimize $\mathcal{N}(\Delta_\sigma, \Delta_\epsilon)$ of Section 2.2. We use $\Lambda = 11$ and the bounding box $[0.510, 0.530] \times [1.30, 1.50]$, i.e. the same range as in Fig. 1. Running our algorithm for 10 different starting points chosen at random within this bounding box, the number $FC$ of function calls to reach a point of negative navigator value was $9 \leqslant FC \leqslant 31$, while $\overline{FC} = 19.3$ on average. All runs terminated at essentially the same point (with an error controlled by $g_{\text{tol}}$)

$$
\begin{aligned}
x_f &= (\Delta_\sigma, \Delta_\epsilon) = (0.5182861212(4), 1.41521640889(6)), \\
N(x_f) &= -0.00267253307546000(2),
\end{aligned}
\tag{5.8}
$$

---

[28]E.g. for Ising and $O(N)$ we can use the fact that they live close to the kink in a single correlator bound, for Ising we can use $c$-minimization [7], OPE scans can be replaced with the cutting surface algorithm [8], etc.

where the tiny error bars show the largest difference observed between different runs. We conclude that the minimum is unique and all the runs terminate near it.

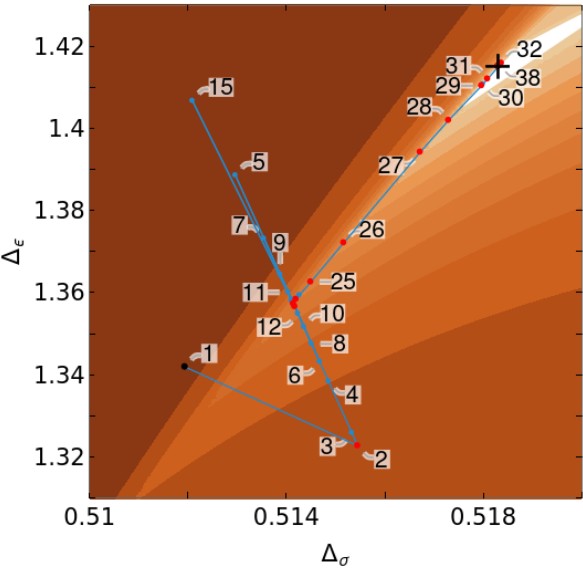

Figure 7: A representative run of our algorithm, see Section 5.3.1. Only the relevant part of the bounding box $[0.510, 0.530] \times [1.30, 1.50]$ is shown. *Black dot:* the initial point $x_0$. *Red dots:* points $x_k$ accepted by the line searches as satisfying the strong Wolfe condition. *Blue dots:* intermediate points where the function was evaluated during the line searches. *Black cross:* position of the found minimum. *Background:* contour plot of the navigator function $\mathcal{N}(\Delta_\sigma, \Delta_\epsilon)$ (darker colors correspond to higher function values, and the white area to the negative navigator, i.e. the Ising island). This run took 29 function evaluations to reach the island, and 66 function evaluations to reach the minimum within the specified $g_{\mathrm{tol}}$ (see Fig. 8). Only the first 38 points are marked, the rest being too closely spaced to be distinguishable.

A representative run is shown in Fig. 7, where the numbered points correspond to the path taken by our modified BFGS algorithm. Convergence rate in this run is illustrated in Fig. 8(left) where we plot the navigator values $\mathcal{N}_i$ returned by subsequent function calls, until the negative navigator region is reached. This plot can be correlated with the navigator shape in Fig. 1, which features an arrow-shaped valley around the Ising island (see Section 3). Thus, Fig. 8(left) shows a period of modest progress in the minimization of $\mathcal{N}(\Delta_\sigma, \Delta_\epsilon)$, in some sense looking for the the valley. This is followed by a period of fast decrease once the valley is found (at around $i = 25$).

Another way to evaluate the convergence rate is shown in Fig. 8(right), where we plot for the same run the distance $\|x_i - x_f\|$ between the point $x_i$ and the eventually found minimum $x_f$. This measure of convergence is appropriate also for the region where $\mathcal{N}(x) < 0$. This plots show a period of greatly accelerated convergence towards

the end of the run. Indeed, we expect Newton-like, i.e. superlinear,[29] convergence in the final stages of the BFGS algorithm. Similar plots for six more runs are collected in App. E.1.[30]

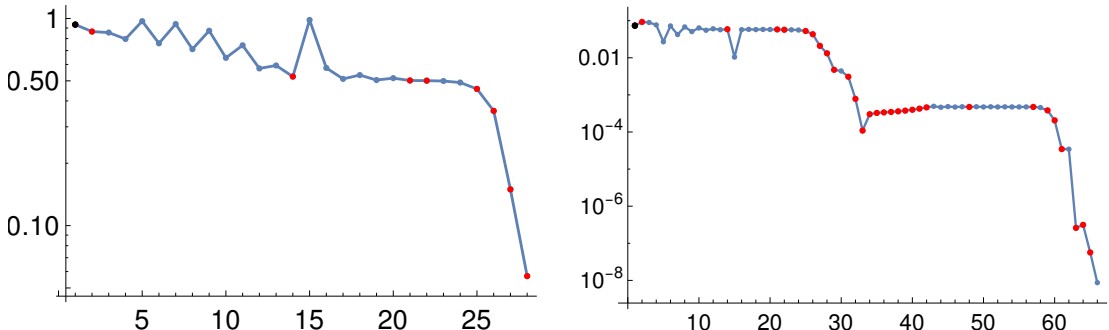

Figure 8:   These plots refer to the run of our modified BFGS algorithm shown in Fig. 7, and use the same color code for the dots. *Left:* Navigator value $\mathcal{N}_i$ at the $i$-th function call. Only the function calls before reaching the negative navigator region are shown in this logarithmic plot. Naturally, function values decrease monotonically along the red dots (points accepted by the line searches), while this condition does not have to hold for the blue dots. *Right:* Logarithmic plot of $\|x_i - x_f\|$ at the $i$-th function call.

Finally, we observe that minimum (5.8) of the $\Lambda = 11$ navigator function gives a good prediction for the actual location of the Ising model, as compared to a generic point in the Ising island. Indeed, the distance between this minimum and the best prediction from [15] (3-parameter scan at $\Lambda = 43$) is only $\sim 10\%$ of the size of the $\Lambda = 11$ island, see Fig. 9.

### 5.3.2   3-parameter searches

We will present next the tests for the three-parameter navigator $\mathcal{N}(\Delta_\sigma, \Delta_\epsilon, \theta)$. We used $\Lambda = 19$, the bounding box $[0.510, 0.530] \times [1.30, 1.50] \times [0.8, 1.1]$, and 20 random initial points within it.

These runs are shown in Fig. 10. Eighteen of them successfully converged to the

---

[29]Recall that superlinear means $\epsilon_{i+1} = o(\epsilon_i)$ where $\epsilon_i$ is the error after step $i$. The Newton method has quadratic convergence, $\epsilon_{i+1} = O(\epsilon_i^2)$, while for the BFGS only weaker theorems showing superlinear convergence are available [21]. One-dimensional bisection in this notation has linear convergence, $\epsilon_{i+1} \leqslant \alpha \epsilon_i$ with $\alpha < 1$.

[30]Once the navigator function is negative, we often observe a bit of a plateau, for example between iterations 30-60 in Fig. 8. At that point we are relatively close to the minimum, but the long sequences of blue dots indicate that the BFGS quadratic model of the navigator function is not yet accurate. It is quite possible that this is caused by the non-$C^2$ locus that we identified above, and it would certainly be interesting to investigate this further. Either way, the algorithm eventually recovers and then continues to converge rapidly to the minimum.

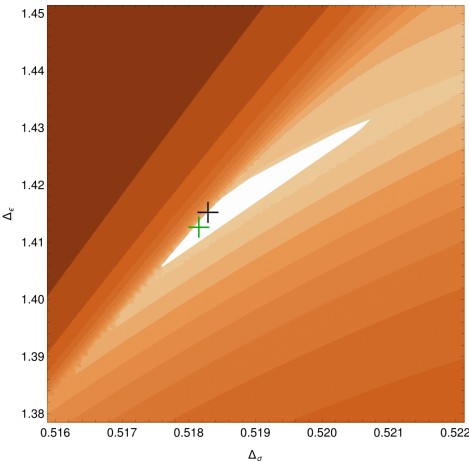

Figure 9: This plot shows that minimum (5.8) of the $\Lambda = 11$ navigator (black cross) is very close to the best available estimate of the true location of the Ising model [15] (green cross), considering the size of the $\Lambda = 11$ Ising island (white region).

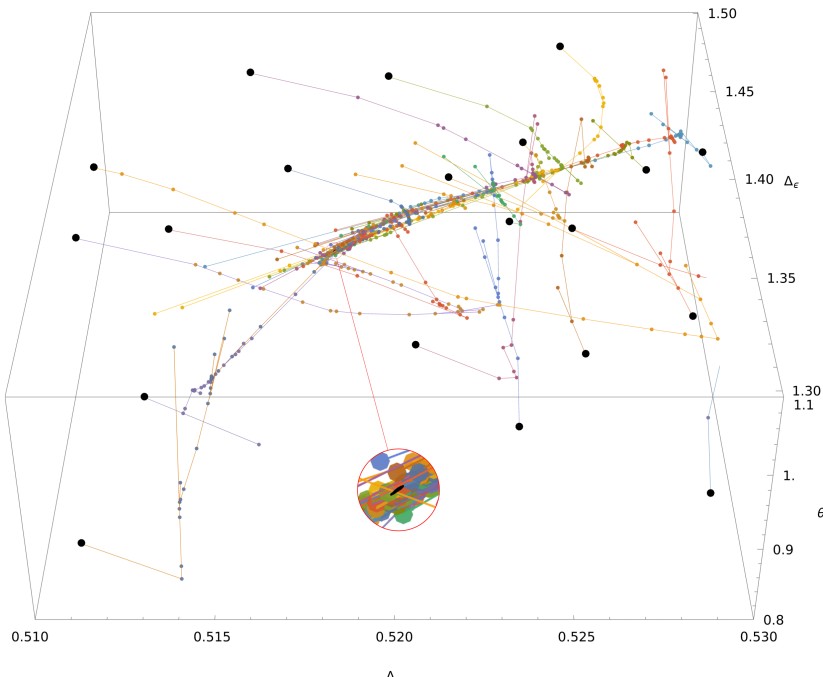

Figure 10: BFGS runs starting at 20 random points from of the bounding box $[0.510, 0.530] \times [1.30, 1.50] \times [0.8, 1.1]$, at $\Lambda = 19$. Initial points are black. Except for two runs that terminated at the boundary (one of them is in the lower right), all the others converged to the same minimum inside the Ising island (see the tiny black shape in the the magnified inset).

same minimum inside the $\Lambda = 19$ Ising island:

$$x_f = (\Delta_\sigma, \Delta_\epsilon, \theta) = (0.5181536110(7), 1.412692879(8), 0.969334757(6)),$$
$$N(x_f) = -0.0000208827730(5). \tag{5.9}$$

A typical successful run is shown separately in Fig. 11.

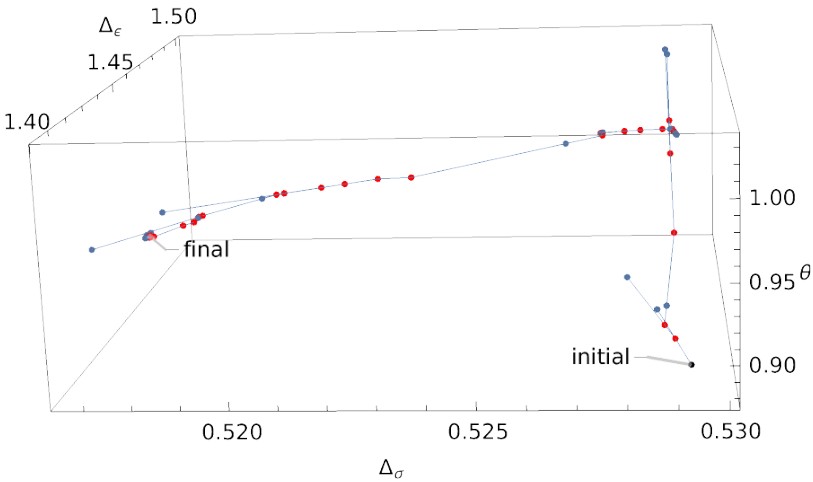

Figure 11:  A typical BFGS run from Fig. 10 (only a part of the bounding box is shown). First, the search is seen to be looking for the "valley", and once it has found it, converges rapidly to the Ising island.

Two runs terminated at the boundary of the bounding box with both the subsequent BFGS search direction and the gradient pointing outside, according to the safe-guarding procedure (see Algorithm 1). This suggests that these points were being attracted by a minimum outside the bounding box. By inspection, these runs started close to the edge of the bounding box in regions where the navigator surface is non-convex even after applying transformation (5.1).

Limiting to the successful runs, it took on average 50.3 function calls to reach the negative navigator region. Of course, the $\Lambda = 19$ island is orders of magnitude smaller in all directions than the bounding box. This demonstrates our point that the navigator minimization method is capable of finding a small isolated island given even a rough estimate of its location. We will comment in Section 5.4 on an iterative way to speed up high-$\Lambda$ calculations.

Using the run in Fig. 11 as an example, we show its rate of convergence in Fig. 12, following the same conventions as in Fig. 8. Comparing Figs. 11 and 12, it's easy to reconstruct what is going on. The initial line searches are spent finding the bottom of the valley. Once this is found, the algorithm quickly manages to follow the valley towards the negative navigator region. Similar plots for six more runs from Fig. 10 are collected in App. E.2.

It's worth pointing out that in both Fig. 8(right) and Fig. 12 we see two periods of accelerated convergence: one when the negative region is approached and another

towards the end of the run. The slower rate of convergence in between might be due to the function exhibiting some local concavity, or due to a large change in the local Hessian. We have not investigated this in detail.

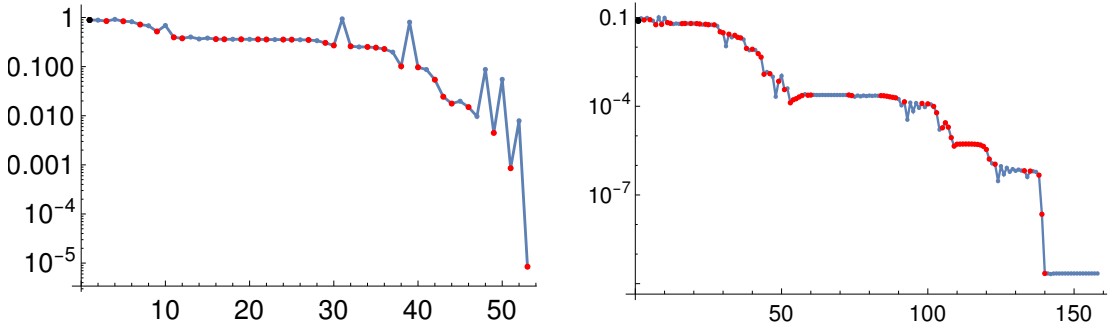

Figure 12: These plots refer to the run of our modified BFGS algorithm shown in Fig. 11, and follow the same convention as Fig. 8, with $\mathcal{N}_i$ on the left and $\|x_i - x_f\|$ on the right.

## 5.4 Other algorithms and possible improvements

We have shown in the previous section that navigator minimization using our modified BFGS algorithm offers a robust and efficient method for finding an allowed point. However, there are bound to be avenues for improvement. We will remark on some potential improvements in this section. We hope that the algorithm presented here sets a good benchmark to which future algorithms will be compared.

In order to efficiently find an allowed point at high values of $\Lambda$, one could imagine an iterative procedure where the navigator minimum point at a lower derivative order is used as an initial point for a minimization run at a higher derivative order (perhaps reducing the bounding box, or inheriting the Hessian estimate from the lower $\Lambda$ BFGS run). This is expected to perform well for two reasons. First, because the navigator minimum provides an excellent estimate of the position of the Ising model, see Section 5.3.1, and hopefully also of other CFTs. Second, because of the accelerated convergence properties of the BFGS algorithm after reaching the convex region around the minimum (see Figs. 8 and 12). Using the exact Hessian computed as explained in Appendix C may be also especially beneficial in the convex region around the minimum.

In the above, we did not make use of the fact that the minimum of the navigator occurs close to $\mathcal{N}(x) = 0$, and that in some cases, one will only be interested in reaching any negative point rather than the minimum. This information could be e.g. incorporated in the initial guess for the Hessian, by scaling the identity matrix such that the initial step aims towards zero of $\mathcal{N}(x)$ in a first-order expansion around the initial point (instead of scaling it so that the initial step explores some percentage of the bounding box, as done here)

As discussed before, we have also found that the navigator function is not globally convex. We have found that in our case, this problem can be mitigated by minimizing

another related, more convex function instead, Eq. (5.1). Even in regions of non-convexity of this transformed function, we have found that line searches provided robustness to the algorithm. Still, other bootstrap problems may require more care when dealing with non-convexity. In such cases, algorithms where the updating formula for the approximate Hessian does not enforce positive-definiteness could be advantageous, see [25].

We have opted in our algorithm to constrain the search space in a rudimentary way via the bounding box, and found this to be adequate for our needs. With that being said, there exist a myriad of other algorithms for constrained optimization that could offer more robustness with the way they deal with constraints. Here we mention two included in SciPy: L-BFGS-B, a bounded limited memory version of the BFGS method optimized for dealing with problems with search spaces with a large number of dimensions, and SLSQP, allowing general, as opposed to box, constraints. See [21] for more information on constrained optimization.

Similarly, there are many unexplored avenues for parallelization. One could imagine parallelizing the line search, or using an inherently parallel optimization algorithm, in the spirit of particle swarms [26]. Particle swarm algorithms that we have seen do not make use of gradient information. Since we have gradients for free (Section 4), it would be desirable to develop a similar algorithm taking advantage of the gradients.

# 6 An application: exploring the tip of an island

In order to connect the Ising island to physical observables it is important to know its extreme points. For example, the left- and rightmost point of the island provide a rigorous lower and upper bound on the critical exponent $\eta = 2\Delta_\sigma + 2 - d$. In previous applications such bounds were often found by simply mapping out the entire island, using a higher-dimensional analogue of a binary search based on a Delaunay triangulation, and then locating its extremal points. A more systematic triangulation algorithm, suitable for parallelization, was introduced in [12] and used to determine the instability of the $O(3)$ fixed point.

In future bootstrap applications one might want to study more complicated systems of correlators and this inevitably means the introduction of new parameters. If we wish to locate the extremal point of an island in such a higher-dimensional space then any triangulation algorithm based on a sequence of feasible and infeasible points will scale extremely poorly. A constrained optimization algorithm based on a navigator function is much less sensitive to the dimensionality of the parameter space and will perform much better. We therefore expect that the use of a navigator function is essential for the high-precision determination of critical exponents in the future.

In the next section we present a simple algorithm inspired by these general ideas. We will then maximize $\Delta_\sigma$ in the Ising island as an illustration.

## 6.1 A constrained optimization algorithm

Suppose we want to locate an extremal point of the allowed region in the direction specified by a vector $n$. The problem is then:

$$\text{maximize } n^T x \qquad \text{over all } x \text{ such that } \mathcal{N}(x) \leqslant 0. \qquad (6.1)$$

We will use optimality conditions

$$\mathcal{N}(x) = 0 \qquad \text{and} \qquad \left( I - \frac{nn^T}{n^T n} \right) \nabla \mathcal{N}(x) = 0, \qquad (6.2)$$

where the latter equation sets to zero all components of the gradient orthogonal to $n$. We propose to work towards a solution of these equations in a manner inspired by the quasi-Newton method from Section 5. We will now explain the full algorithm (see Algorithm 2 below for a summary).

As in Section 5 we will use a quadratic model around a point $x_k$:

$$\mathcal{N}(x) \approx \mathcal{N}^{(2)}(x) := \mathcal{N}(x_k) + \nabla \mathcal{N}(x_k)(x - x_k) + \frac{1}{2}(x - x_k)^T B_k (x - x_k). \qquad (6.3)$$

The function and the gradient at $x_k$ are assumed known, while $B_k$ can be either the exact Hessian at $x_k$ (computed as explained in Appendix C), or an approximation like the one obtained from the BFGS method. In the following we will assume that $B_k \succ 0$.

Substituting the quadratic model in (6.2) we find a simple system involving one quadratic and many linear equations, which can be solved exactly, yielding two solutions.[31] These are real if $x_k$ in the allowed region, so that $\mathcal{N}(x_k) < 0$, and by continuity also in some domain outside the feasible region. In this case the surface $\mathcal{N}^{(2)}(x) = 0$ is an ellipsoid, and the second condition in (6.2) picks out the extremal points of this ellipsoid along the $n$ direction. Some distance away from the allowed region the ellipsoid shrinks to zero size and the solutions become complex-conjugate. We denote by $x_\#$ the real solution which has the largest value of $n^T x$, when the solutions are real. When the solutions are complex conjugate, we let $x_\#$ denote their real part (and then $x_\#$ turns out to simply correspond to the minimum of the model function).

Denote $p_k = x_\# - x_k$; this is our search direction. The next point $x_{k+1}$ is then found using a line search along $p_k$ starting from $x_k$. We use the initial step length $\alpha = 1$, however the rest of the line search algorithm is not the Moré-Thuente algorithm used in BFGS. This should not be surprising since we are now solving a different problem. Instead of minimizing $\mathcal{N}(x)$ we would now like to maximize $n^T x$ while moving along a trajectory remaining close to the boundary of the allowed region (but not exactly along the boundary). One could think that a safe choice would be to remain always inside

---

[31]This is where our algorithm differs significantly from conventional constrained optimization algorithms like sequential quadratic programming methods or interior point methods (see e.g. [21]). The latter solve a linear system at each step in order to be applicable very generally. Such a linearization is unnecessary here because we only have a single quadratic equation.

the allowed region (a sort of interior point algorithm). We have found however that a much faster algorithm results if we allow the algorithm to choose points on both sides of the boundary. To make sure that the algorithm does not veer off too much away from the boundary, we impose

$$\mathcal{N}(x_{k+1}) \leqslant \lambda_{\text{rel}}|\mathcal{N}(x_k)| \tag{6.4}$$

with a parameter $\lambda_{\text{rel}} > 0$. Clearly $\lambda_{\text{rel}} < 1$ would be safer but might slow down the algorithm in the later stages. We found it advantageous to use $\lambda_{\text{rel}}$ somewhat above 1, e.g. $\lambda_{\text{rel}} = 2$ works well.

So (6.4) is our line search termination condition. In practice, this condition with $\lambda_{\text{rel}} = 2$ is not very constraining and the initial step $\alpha = 1$ is almost always accepted if we start with a good initial Hessian. (E.g. in the run shown in Section 6.2 this happened for 100% of the steps.) In the cases that the initial step $\alpha = 1$ does not obey Eq. (6.4), we proceed as follows. We construct cubic polynomial approximation $P(\alpha)$, fitted to match the value and gradient at the initial point and the previous line search point. If $x_k$ is in the feasible region we choose the next $\alpha$ by solving $P(\alpha) = 0$, and if not then by minimizing $P(\alpha)$. Iterating this, eventually we find an $\alpha$ such that $x_{k+1} = x_k + \alpha p_k$ satisfies (6.4).

Once we have accepted $x_{k+1}$, we construct a new quadratic model around this point. In particular, if the approximate Hessian is used, then $B_k$ is updated as in BFGS. However, the update is carried out only if the curvature condition is obeyed at $x_{k+1}$; as explained in footnote 24 this is sufficient to ensure that $B_{k+1} \succ 0$. If the curvature condition is not satisfied, then the Hessian is not updated.

We then repeat the process. The algorithm terminates if the conditions (6.2) are obeyed within a certain tolerance.

## 6.2 The tip of the Ising island

As an example, let's apply the above algorithm to find the maximal value of $\Delta_\sigma$ within the Ising allowed island $\mathcal{N}(\Delta_\sigma, \Delta_\epsilon) \leqslant 0$ where $\mathcal{N}$ is the 2-parameter navigator for the Ising 3-correlator setup at derivative order $\Lambda = 11$.[32] The search was started from the navigator minimum reached via a BFGS run, and with the initial Hessian approximation $B_0$ inherited from BFGS, which is expected to be close to the true Hessian. The algorithm path is shown in Fig. 13. The algorithm took 17 steps to reach the tip of the island, i.e. the point with maximal $\Delta_\sigma$. Termination condition $\max(|\mathcal{N}(\Delta_\sigma, \Delta_\epsilon)|, |\partial_{\Delta_\epsilon}\mathcal{N}(\Delta_\sigma, \Delta_\epsilon)|) \leqslant g_{\text{tol}}$ was satisfied with $g_{\text{tol}} = 10^{-27}$.

For comparison, Fig. 13 also show the blue allowed region obtained from the Delaunay triangulation method. We finely sampled the zoomed-in region around the very tip

---

[32]In this test, unlike in Section 3, we have not imposed the OPE relation $\lambda_{\sigma\sigma\epsilon} = \lambda_{\sigma\epsilon\sigma}$, i.e. the navigator was defined imposing positivity separately on the two terms in (2.29), which is precisely the setup in [14]. There is no particular reason for this difference with Section 3.

**Input:** A navigator function $\mathcal{N}(x)$, a vector $n$ indicating the maximizing
   direction, a precision goal $g_{\text{tol}}$ and a line search parameter $\lambda_{\text{rel}}$.
**Output:** The final point $x_f$.
**begin**

   Use Algorithm 1 to construct a feasible point $x_0$ and Hessian estimate $B_0$

   $x_{\text{lastBFGS}} = x_0$

   $B_{\text{lastBFGS}} = B_0$

   **while** $\| \left( I - (n^T n)^{-1} n n^T \right) \nabla \mathcal{N}(x_k)\| > g_{\text{tol}}$ *or* $|\mathcal{N}(x_k)| > g_{\text{tol}}$ **do**

      $p_k = \text{search\_direction}(x_k, n, \mathcal{N}(x_k), \nabla\mathcal{N}(x_k), B_k)$

      $\alpha = 1$

      **while** $\mathcal{N}(x_k + \alpha p_k) > \lambda_{\text{rel}}|\mathcal{N}(x_k)|$ **do**

         $P(\alpha)$ is interpolating polynomial obtained from $\mathcal{N}(x_k)$, $\mathcal{N}(x_k + \alpha p_k)$
         and their gradients

         **if** $\mathcal{N}(x_k) < 0$ **then**

            find $\alpha$ such that $P(\alpha) = 0$

         **else**

            find $\alpha$ such that $P(\alpha)$ is minimized

         **end**

      **end**

      $x_{k+1} = x_k + \alpha p_k$

      **if** $(x_{k+1} - x_{\text{lastBFGS}})^T(\nabla\mathcal{N}(x_{k+1}) - \nabla\mathcal{N}(x_{\text{lastBFGS}})) > 0$ **then**

         $B_{k+1} = \text{BFGS\_update}(x_k, B_k; x_{\text{lastBFGS}}, B_{\text{lastBFGS}})$

         $x_{\text{lastBFGS}} = x_{k+1}$

         $B_{\text{lastBFGS}} = B_{k+1}$

      **else**

         $B_{k+1} = B_k$

      **end**

   **end**

   Return $x_k$.

**end**

Algorithm 2: An algorithm for finding the extremal point of an island.

of the island, with a total number of sampled points being around 480 [33]. In contrast, our algorithm takes only 10 steps to locate the maximal $\Delta_\sigma$ point more accurately than the triangulation resolution. The line search never had to be activated, the initial try $\alpha = 1$ having been accepted in 100% of the steps.

In Fig. 14 we show the convergence rate towards the minimum. These plots demonstrate superlinear convergence towards the end of the run, as should be expected from this type of algorithm.

We would like to warn the reader about a difference in spirit between our Algorithms 1 and 2. Algorithm 1 for navigator minimization is backed up by decades of experience in numerical optimization, and should be widely applicable without major modifications. On the other hand, Algorithm 2 is our own custom-made procedure. It served well the purpose to demonstrate the point that the navigator can be used to find extremal values of allowed parameters, but it has a somewhat tentative character and is expected to evolve more in the future.

For example, the Ising island is admittedly a simple model with a convex island and a single local maximum of $\Delta_\sigma$. If the island does not have such a nice shape, Algorithm 2 can get stuck in a local optimum instead of the global optimum. In more realistic cases it is therefore important to have a rough idea of the shape of the island, and then perhaps an admixture of triangulation-based methods and navigator methods might be the best approach.

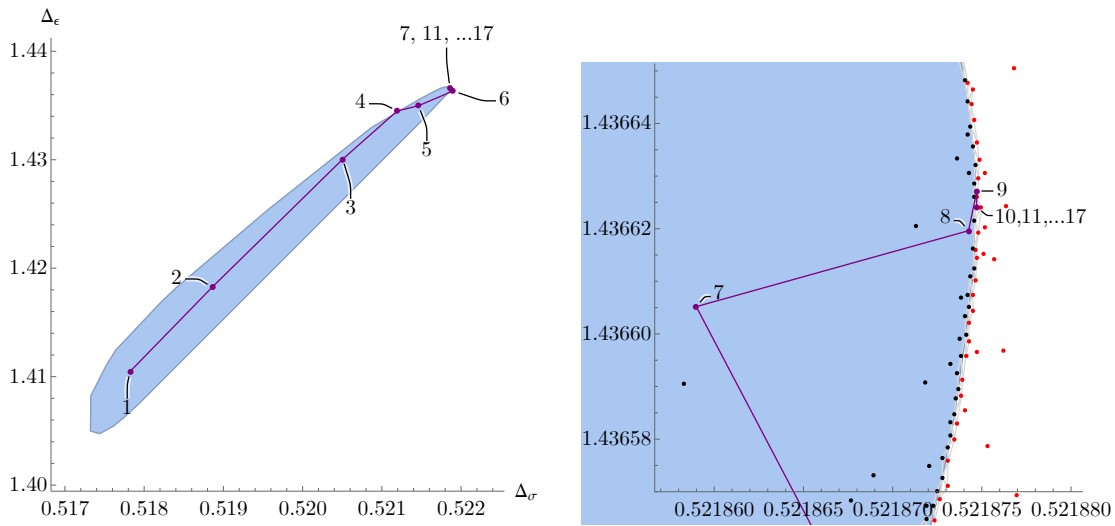

Figure 13: *Magenta path*: A run of Algorithm 2, see the text, with a zoom-in on the right. *Blue region*: the Ising island from the Delaunay method with black/red being the allowed/excluded points.

---

[33]To test the feasibility of those points, we only require SDPB to find primal/dual jumps. In general such a run is quicker than a typical SDPB run for an optimal solution. However, in terms of total number of SDPB iterations, we find that the 480 feasibility runs correspond to around 4705 SDPB iterations, while the 17 optimal runs correspond to around 1080 SDPB iterations. We still conclude that our method has a significant advantage.

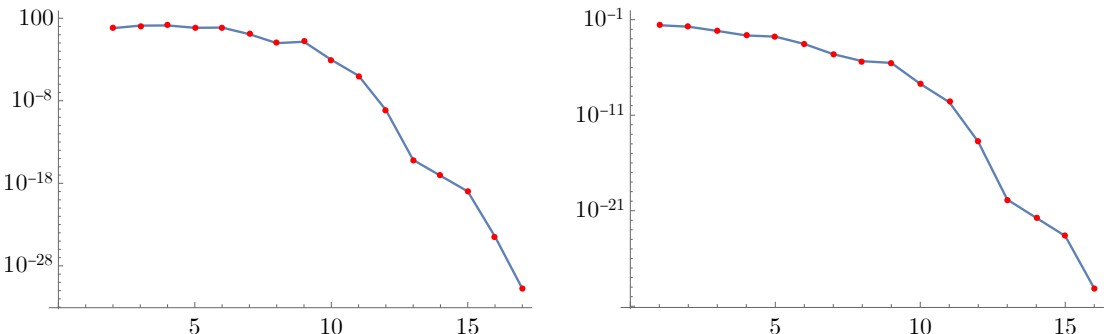

Figure 14: These plots refer to the run of our Constrained BFGS algorithm shown in Fig. 13. *Left:* Logarithmic plot of $|\partial_{\Delta_\epsilon}\mathcal{N}(\Delta_\sigma, \Delta_\epsilon)|$ at the $i$-th function call. *Right:* Logarithmic plot of $\|x_i - x_f\|$ at the $i$-th function call. Note that line search never had to be activated, as the initial step $\alpha = 1$ always satisfied condition (6.4) with $\lambda_{\text{rel}} = 2$ used in this run.

# 7 Conclusions and future directions

We have presented in this work a powerful alternative to the scanning-based approach employed so far in the numerical conformal bootstrap program. This came from the realization that there exist functions, for which we have coined the term "*navigator functions*," which measure how far a given point is from the boundary between allowed and disallowed regions and can thus be used to efficiently find an allowed point as well as the boundary of an allowed region. We have explicitly constructed two such navigator functions. It was shown that the computation of these navigator functions can be written as a semi-definite programming problem of the same form as an OPE maximization. Adding the generalized free field solution to the crossing equation has led us in Section 2.1.1 to the definition of the GFF navigator. The $\Sigma$-navigator was introduced in Section 2.1.2 as an another equally valid option.[34]

With the help of such functions, we have shown it is possible to quickly locate allowed regions in parameter space by ways of minimization. We have presented in Algorithm 1 a modified BFGS algorithm which does so quite efficiently. To prove this, we set out to study the canonical bootstrap problem of the 3d Ising model. First we showed that the navigator is $(C^1)$ smooth and has no local minima in the disallowed region. With both a two-dimensional search space at $\Lambda = 11$, and a three-dimensional search space at $\Lambda = 19$, we have shown that it took on average a few dozen SDPB calls to find the Ising island (19.3 for the former, 50.3 for the latter), starting only from very

---

[34]While the GFF-navigator is naturally normalized, the $\Sigma$-navigator has its own set of advantages. It is actually easier to set up, since one does not have to work out the GFF OPE coefficients. In addition, there is not one but infinitely many $\Sigma$-navigators, corresponding to different choices of terms in the r.h.s. of (2.7) or (2.25), and this flexibility may prove useful in the future. At present, we see no definite reason to prefer one or the other navigator. For comparison, we performed some of the reported computations using both navigators (e.g. section 6.2), and they performed equally well.

conservative estimates of the parameters. This is competitive with previous methods for isolating islands and bounding CFT data. Moreover, these previous methods suffered from exponential scaling with the dimensionality of the search space. This constituted a major bottleneck for the kinds of problems that could be tackled: realistically only setups with a handful of free parameters could be considered. We expect that the scaling of the minimization-based navigator method with the number of parameters will greatly outperform scanning methods.

Crucially, efficient minimization of a navigator function, for example with the BFGS algorithm presented in this paper, requires the knowledge of derivatives of the navigator function. We have derived the "SDP gradient formula," Eq. (4.16), which gives the variation of the objective function of an SDP as only a function of the variation of the SDP input parameters around the point where the derivative is requested. This means that computing derivatives does not require additional SDPB runs, making one function *and* gradient evaluation in a BFGS run just about equivalent in cost to one OPE maximization.

We also tested the efficiency of the navigator method to search for extremal parameter values allowed by the bootstrap constraints. So, we presented in Section 6 a way to find optimal bounds on CFT data using a custom-made constrained-optimization routine. The algorithm was able to walk in and around the allowed region and converge in 17 steps to the maximal allowed $\Delta_\sigma$, determining it to an accuracy of $\sim 10^{-35}$.[35] A similar triangulation-based search only achieves an accuracy of $10^{-6}$ even after testing over 400 points, see Fig. 14. Again we expect that the increase in performance can only become greater as the dimensionality of the search space increases.

We feel that the applications shown in this paper demonstrate only a small part of the power the navigator method, and we are hopeful that the future will show it to be a great addition to the toolbox of all bootstrap enthusiasts.

We would like to conclude by mentioning here some of the ideas that we are going to start exploring immediately ourselves using this new tool. Indeed, these applications, out of reach of traditional bootstrap techniques, were among our chief motivations to start thinking hard about the navigator function.

One class of situations where navigator is going to be useful is when we know a solution to bootstrap constraints for some value of a parameter (such as space dimension $d$ or the symmetry group rank $N$) and we would like to perform a deformation in this parameter. We imagine doing this by considering a navigator function depending on the dimensions of several exchanged operators, and imposing sparsity of the exchanged spectrum. Among other things, this should allow a more robust determination of critical values of parameters when bootstrap solutions disappear, than the more traditional approach of looking for kinks and trying to see when those kinks get rounded off. One long-standing problem which could benefit from this approach is determining the upper critical dimension of the 3-state Potts model. Including exchanged operator dimensions

---

[35]The order of magnitude for the difference of last two points in $\Delta_\sigma$ is around $10^{-35}$. Another estimation is that $\mathcal{N}(x)/\|\nabla\mathcal{N}(x)\|$ for the last point is around $10^{-37}$.

among the arguments of the navigator function could also provide a useful (and more rigorous) alternative to estimating the spectrum using the extremal functional method [9, 27, 7].

The use of the navigator function to quickly find extremal allowed values (Section 6) will benefit all cutting-edge bootstrap computations. One problem on our to-do list is to bootstrap the system of correlators in O(3) symmetric CFTs involving lowest scalar primaries in vector ($\phi$), scalar ($s$), rank-2 tensor ($t$), and rank-4 tensor ($t_4$) O(3) representations. This setup extends that of [12] by including $t_4$ as an external operator. The physics interest in doing so is that it will allow access to the OPE coefficient $\lambda_{t_4,t_4,t_4}$, and other data needed to study the RG flow leading from the O(3) fixed point to the cubic fixed point in conformal perturbation theory (see [12], Section 5). The parameter space for this problem is 13-dimensional (4 $\Delta$'s and 9 OPE coefficients), out of reach of traditional approaches, but we expect that the navigator function will put it within reach.

As a final example, we expect that the navigator functions will allow an exploration of hybrid methods where the numerical bootstrap data is complemented with analytical data at high spins obtained from the light-cone bootstrap, as suggested in Section 9.1 of [28]. We imagine a navigator function depending on many parameters accurately parametrizing one or more Regge trajectories. In this context a navigator function will be very useful not only to localize an allowed point, but also because the minimum of the navigator offers a natural "most feasible point" that can be used to compare different parametrizations. Although this method is not entirely rigorous, it might lead to more precise estimates of the numerical bounds.

# Acknowledgements

We thank Tom Hartman for important conversations that sparked this exploration. We thank Walter Landry for discussions and for collaboration on the program `approx_objective` for computing variations of the objective function. NS thanks Shixin Zhang, Yinchen He for inspiring discussions. NS thanks his parents for support during the COVID-19 pandemic.

MR is supported by Mitsubishi Heavy Industries (MHI-ENS Chair). BS is supported by a *Fonds de Recherche du Québec – Nature et technologies* B1X Master's scholarship. DSD is supported by Simons Foundation grant #488657 (Simons Collaboration on the Nonperturbative Bootstrap) and a DOE Early Career Award under grant no. DE-SC0019085. BvR is supported by Simons Foundation grant #488659 (Simons Collaboration on the Nonperturbative bootstrap). NS is supported by European Research Council (ERC) under the European Union's Horizon 2020 research and innovation programme (grant agreement no. 758903). SR is supported by the Simons Foundation grant 488655 and 733758 (Simons Collaboration on the Nonperturbative Bootstrap), and by Mitsubishi Heavy Industries as an ENS-MHI Chair holder.

Some of the computations in this work were performed on the Caltech High Per-

formance Cluster, partially supported by a grant from the Gordon and Betty Moore Foundation. This work also used the Extreme Science and Engineering Discovery Environment (XSEDE) Comet Cluster at the San Diego Supercomputing Center (SDSC) through allocation PHY190023, which is supported by National Science Foundation grant number ACI-1548562. The computations in this paper were partially run on the Symmetry cluster of Perimeter institute and on the Hopper cluster of the École Polytechnique.

# A    Tweaks of the GFF-navigator

As mentioned in Section 2.1.1 and footnote 12, the GFF-navigator definition has to be tweaked in presence of additional GFF operators violating gap assumptions. These modifications will be discussed here. In addition we will explain how to deal with the case where the navigator function depends on the magnitude of a squared OPE coefficient.

A relevant example in the single-correlator setup of Section 2.1.1 is to assume a gap in the scalar spectrum above $\Delta_*$. E.g. suppose that all further scalars above the one at $\Delta_*$ are required to be above $\Delta_{\text{gap}}$. This corresponds to changing the constraint $\Delta \geqslant \Delta_*$ for $\ell = 0$ in (2.2) to "$\Delta = \Delta_*$ or $\Delta \geqslant \Delta_{\text{gap}}$." We can still define the navigator by the same Eq. (2.5). In this case we don't in general expect the navigator to be monotonic in the $\Delta_*$ direction. For large $\Delta_{\text{gap}}$, definition (2.6) of $M_{\text{GFF}}$ will have to be modified, including all scalar GFF conformal block contributions below $\Delta_{\text{gap}}$:

$$M_{\text{GFF}}(u,v) = \sum_{n \geqslant 0\,:\, 2\Delta_\phi + 2n \leqslant \Delta_{\text{gap}}} c_n F_{2\Delta_\phi + 2n, 0}(u,v), \tag{A.1}$$

where $c_n$ are explicitly known coefficients ($c_0 = 2$). These are contributions of GFF operators of schematic form $\phi \Box^n \phi$.

For the 3-correlator setup, let us discuss how the GFF-navigator definition (2.23) should be modified in the case of gap assumptions in the spectrum of $\ell \geqslant 1$ operators. As a concrete example, let us define the navigator $\mathcal{N}(\Delta_\sigma, \Delta_\epsilon, c_T)$ where $c_T$ is the 2pt function coefficient of the canonically normalized stress-tensor. The $c_T$ parametrizes the OPE coefficients of the corresponding unit-normalized $\Delta = 3$, $\ell = 2$ primary $\mathcal{O}^+$ as:

$$\lambda_{\sigma\sigma\mathcal{O}} = K_3 \frac{\Delta_\sigma}{\sqrt{c_T}}, \qquad \lambda_{\epsilon\epsilon\mathcal{O}} = K_3 \frac{\Delta_\epsilon}{\sqrt{c_T}} \tag{A.2}$$

where $K_d$ is a known $d$-dependent constant. To isolate the stress tensor, we need to impose a gap assumption on the higher-dimension $\ell = 2$ $\mathcal{O}^+$ operators. We will assume that all of them have $\Delta \geqslant \Delta_{\text{gap}}$ where $\Delta_{\text{gap}} > 3$ is some fixed parameter. E.g. let us choose $\Delta_{\text{gap}} = 5$, which allows the 3d Ising CFT.[36]

---

[36]Recall that the 3d Ising CFT has $\Delta_{T'} = 5.50915(44)$ [28].

For this problem, the analogue of Eq. (2.21) will be

$$
\vec{V}_{0,0} + \lambda \vec{M} + \mathrm{Tr}\left[ P_{\Delta_\epsilon,0}\left(\vec{V}_{+,\Delta_\epsilon,0} + \begin{pmatrix} 1 & 0 \\ 0 & 0 \end{pmatrix}\vec{V}_{-,\Delta_\sigma,0}\right)\right]
$$
$$
+ \frac{(K_3)^2}{c_T}\,\mathrm{Tr}\left[\begin{pmatrix} \Delta_\sigma^2 & \Delta_\sigma\Delta_\epsilon \\ \Delta_\sigma\Delta_\epsilon & \Delta_\epsilon^2 \end{pmatrix}\vec{V}_{+,3,2}\right]
$$
$$
+ \sum_{(\Delta,\ell)\in S_+}\mathrm{Tr}\left[P_{\Delta,\ell}\vec{V}_{+,\Delta,\ell}\right] + \sum_{(\Delta,\ell)\in S_-} p_{\Delta,\ell}\vec{V}_{-,\Delta,\ell} = 0\,, \quad \text{(A.3)}
$$

where the stress tensor contribution is now isolated, and $S_+$ compared to (2.18) implements the stronger requirement that $\Delta \geqslant \Delta_{\mathrm{gap}}$ for $\ell = 2$.

To define the GFF navigator, we will proceed analogously to (A.1) and include in $\vec{M}_{\mathrm{GFF}}$ additional terms corresponding to all GFF primaries violating the gap assumptions. In the case at hand, we have to check the spin-2 GFF operators of the schematic form $\sigma\partial\partial\Box^n\sigma$ and $\epsilon\partial\partial\Box^n\epsilon$. For $\Delta_{\mathrm{gap}} = 5$ and $\Delta_\sigma, \Delta_\epsilon$ around the 3d Ising island, only the $n = 0$ operators of this form are below the gap. So we take

$$
\vec{M}_{\mathrm{GFF}} = \mathrm{Tr}\left[\begin{pmatrix} 2 & 0 \\ 0 & 0 \end{pmatrix}\vec{V}_{+,2\Delta_\sigma,0}\right] + \mathrm{Tr}\left[\begin{pmatrix} c(\Delta_\sigma) & 0 \\ 0 & 0 \end{pmatrix}\vec{V}_{+,2\Delta_\sigma+2,2}\right]
$$
$$
+ \mathrm{Tr}\left[\begin{pmatrix} 0 & 0 \\ 0 & 2 \end{pmatrix}\vec{V}_{+,2\Delta_\epsilon,0}\right] + \mathrm{Tr}\left[\begin{pmatrix} 0 & 0 \\ 0 & c(\Delta_\epsilon) \end{pmatrix}\vec{V}_{+,2\Delta_\epsilon+2,2}\right] + \vec{V}_{-,\Delta_\sigma+\Delta_\epsilon,0}
$$
$$
- \frac{(K_3)^2}{c_T}\,\mathrm{Tr}\left[\begin{pmatrix} \Delta_\sigma^2 & \Delta_\sigma\Delta_\epsilon \\ \Delta_\sigma\Delta_\epsilon & \Delta_\epsilon^2 \end{pmatrix}\vec{V}_{+,3,2}\right], \quad \text{(A.4)}
$$

where $c(\Delta_\phi)$ is the (explicitly known) coefficient of the $\Delta = 2\Delta_\phi + 2$, $\ell = 2$ conformal block in the decomposition of the GFF 4pt function $\langle\phi\phi\phi\phi\rangle$.

The first two lines in (A.4) are the analogue of (A.1). The last line is an additional small modification needed due to the presence of OPE coefficient parameter $c_T$ among navigator function variables. It is the negative of the stress tensor contribution in (A.3). Including this piece into $\vec{M}_{\mathrm{GFF}}$ is needed to guarantee that problem (A.3) has a solution with $\lambda = 1$ for any fixed value of $c_T$. This in turn guarantees that the navigator $\mathcal{N}(\Delta_\sigma, \Delta_\epsilon, c_T)$ is bounded from above by 1 for any value of its arguments.

# B  Feasibility as optimization

In this appendix we discuss the problem of finding a navigator function from the more abstract semidefinite programming perspective. We will assume the reader is familiar with the semidefinite programming terminology of Section 4.1 in the main text. As we review there, a general numerical bootstrap problem of the opimization type can be formulated as the dual problem given in Eq. (4.1) on p.20. For a feasibility problem, on the other hand, the question is merely whether there exist any $y$ and $Y$ that obey the constraints. In that case the standard approach is to set $b = 0$ in (4.1) and run SDPB until one of two termination conditions are met:

- If a dual feasible point $(y, Y)$ is found, terminate with 'success';

- If a primal feasible point $x$ is found and $c^T x < 0$, then terminate with 'failure'.

The last termination condition is explained by the duality gap: if $b = 0$ then $D(x, y) = c^T x$, which can only be negative (for a primal feasible $x$) if no dual feasible point exists.[37]

The above two termination conditions correspond to the binary oracle output discussed in Section 1: "success" means that the point is excluded (CFT does not exist), while "failure" means that the point is allowed (CFT may exist).

To pass from this to a navigator function, we need to reformulate the feasibility search as an optimization problem. The commonly adopted approach to do so is to use *slack variables* that relax the constraints. As discussed in the main text, in the context of the conformal bootstrap one can add an additional term to the crossing equations, in such a way that these equations can always be obeyed if the coefficient of this extra term is positive. The minimization of the coefficient of this term is then a potential navigator function: if it is positive we are in the 'success' region and if negative we are in the 'failure' region.

We will now describe an alternative navigator function construction, which does not rely on the physical intuition of the crossing symmetry equations. Instead, we will start with a general feasibility semidefinite program of the type (4.1) with $b = 0$, and transform it into an optimization SDP.

As a first attempt, consider replacing the condition $Y \succeq 0$ in (4.1) with a maximization problem:

$$Y \succeq 0 \qquad \implies \qquad \text{maximize } \nu \in \mathbb{R} \text{ such that } Y - \nu I \succeq 0 \qquad \text{(B.1)}$$

with $I$ the identity matrix. With this transformation the 'success' and 'failure' cases mentioned above respectively correspond to $\nu > 0$ and $\nu < 0$ at optimality, and (in the conventions of the main text) we can take $\nu$ at optimality as a candidate navigator function.

Unfortunately the modification (B.1) is not guaranteed to give a finite navigator in the "success" region. E.g. suppose there exists a $Y' \succ 0$ such that $\text{Tr}(A_* Y') = 0$. In the 'success' region one can add this $Y'$ to any feasible solution $Y$ with arbitrarily large coefficient. This would then imply that $\nu \to +\infty$ at optimality. We therefore cannot exclude a divergence in this candidate navigator function unless we know that the program does not allow such $Y'$.

To guarantee boundedness in the 'success' region, we apply the same idea, but on the primal side, that its, by modifying the primal problem (4.2). For simplicity, let us

---

[37]With $b = 0$ the primal problem is completely homogeneous in the sense that the constraints are invariant under rescalings $x \to \lambda x$ with non-negative $\lambda$. In particular, there is an obviously primal feasible point $x = 0$. Since this point teaches us nothing about dual feasibility, the inequality in the second termination condition has to be strict. Furthermore, if we were to ignore the above termination conditions and run the program to optimality then we would either find $x \to 0$ (in the 'success' case) or $x$ diverges such that $c^T x \to -\infty$ (in the 'failure' case). We thank Petr Kravchuk for a discussion of these issues.

first assume that there always exists an $x$ such that

$$B^T x = 0, \qquad c^T x < 0,$$
(B.2)

meaning we only need to introduce a slack variable for the positive semidefiniteness condition. In that case the right problem to solve is:

$$
\begin{aligned}
&\text{minimize } \nu \quad \text{over} \quad x \in \mathbb{R}^P, \nu \in \mathbb{R} \\
&\text{such that } X(x) := x^T A_* + \nu I \succeq 0 \\
&\qquad\qquad B^T x = 0 \\
&\qquad\qquad c^T x = -1
\end{aligned}
$$
(B.3)

This is a standard primal semidefinite programming problem, and we can run it to optimality without special termination conditions. The value of $\nu$ at optimality is the navigator function. In the 'success' region it is guaranteed to be positive (and finite) and then it is likely to be as good a navigator function as the ones used in the main text.

The dual version of the program in (B.3) is:

$$
\begin{aligned}
&\text{maximize } -\xi \quad \text{over} \quad y \in \mathbb{R}^n, Y \in \mathcal{S}^K, \xi \in \mathbb{R} \\
&\text{such that } Y \succeq 0 \\
&\qquad\qquad -c\xi = By + \text{Tr}(A_* Y) \\
&\qquad\qquad \text{Tr}(Y) = 1
\end{aligned}
$$
(B.4)

As usual, the introduction of free variables on one side yields additional constraints on the other side. In this case the trace condition on $Y$ guarantees the boundedness of the problem, and the parameter $\xi$ allows for the re-scaling of a feasible $(y, Y)$ such that this constraint can be met.

Let us also discuss boundedness (from below) in the 'failure' region of (B.3). We do not have a first-principles argument for boundedness everywhere:[38] for the same reasons as above, the navigator function of (B.3) diverges in the 'failure' region if there exists a $x'$ which obeys

$$(x')^T A_* \succ 0, \qquad B^T x' = 0, \qquad c^T x' = 0.$$
(B.5)

Fortunately, in conformal bootstrap applications this is unlikely. To see this, recall that the formulation (4.1) with $c$ and $b$ arises only after eliminating one component of $y$ from a normalization condition $n^T y = 1$ for some normalization vector $n$, which is typically the identity operator. Reinstating this normalization condition as a separate constraint to (4.1) one finds that unboundedness of the modification (B.4) can really only occur

---

[38]Of course the problem becomes trivially bounded if we impose that $\nu > -1$ in the primal problem. This is however all but guaranteed to result in a non-smooth (and locally constant) navigator function in the primal feasible region, which is of limited use for our purposes.

if there is a solution to the crossing symmetry equations (with positive coefficients) without an identity operator. Although this is known to be the case for problems in $d = 2$ and $d = 1$, it is an unlikely possibility in most numerical bootstrap problems and then (B.4) is also bounded in the 'failure' region. The corresponding navigator therefore obeys the same manifest properties as those used in the main text.

Finally let us consider the case where the equality constraints in the primal problem cannot obviously be met. In that case not all is lost: one can simply replace them with

$$B^T x = b + \nu \mathbf{1} - \lambda, \qquad \lambda > 0, \qquad (B.6)$$

with $\mathbf{1} = (1, 1, \ldots 1)$ a constant vector, and proceed by minimizing $\nu + \sum_i \lambda_i$. As before, a positive value at optimality means that no feasible point exists and so we still have a good candidate for a navigator function in the 'success' region.

The navigator functions introduced in this appendix are more general since they work for any feasibility problem of the type described in Eq. (4.1) with $b = 0$. On the other hand, for numerical conformal bootstrap applications they offer little upside compared to the GFF and $\Sigma$-navigators discussed in the main text. Furthermore they also suffer from a practical disadvantage. To see this, note that the GFF and $\Sigma$-navigators are readily implemented with the usual conformal bootstrap software: programs like `sdp2input` or `pvm2sdp` can be used to translate the problems into a format acceptable by SDPB, which e.g. involves setting up matrices $B$ and $A_*$, and SDPB then does the rest of the computation. Unfortunately this workflow does not quite work for the navigator function described in Eq. (B.4). The main problem is that SDPB is meant to solve problems where the matrices $A_p$ have rank one and the constraint $\text{Tr}(Y) = 1$ is not of this form.[39]

## C  Comments on variations of the objective

In section 4, we found a simple formula (4.16) for the linear-order variation in the objective function under changing the SDP. In this appendix, we give a formula for the quadratic-order variation as well, and explain how it can be computed easily using machinery already present in SDPB. We also present numerical checks of both the linear and quadratic variations, determining how their errors scale with the duality gap.[40]

---

[39]One can probably impose the trace constraint in an SDPB compatible way, by extending $y$ with spurious variables $\hat{y}$. One then needs to set these equal to the diagonal components of $Y$ in the sense that $\hat{y}_1 = Y_{11}$, $\hat{y}_2 = Y_{22}$, etc. This can be done by including one additional equation for each diagonal value of $Y$ by extending $b$, $c$, $B$ and $A$. Finally, by extending these quantities by one more entry we can impose the trace constraint by demanding $\sum_i \hat{y}_i = 1$. Alternatively one can use this equation to eliminate one of these extra components instead. It is unclear whether such an altered semi-definite problem still corresponds to any polynomial matrix problem.

[40]The quadratic variation of the objective could be used to compute the Hessian of the navigator function, enabling the use of Newton's method for finding allowed points and extremizing CFT data. We leave possible applications of the quadratic variation to future work.

## C.1 A formula for the quadratic variation

Consider changing an SDP by $(b, c, B, A) \to (b, c, B, A) + (db, dc, dB, dA)$. For simplicity, we assume $dA = 0$. (In practice, we can ensure this by keeping constant the "bilinear basis" and "sample scalings" discussed in [5].) The linear-order change in the objective at optimality is

$$dL = db^T y + dc^T x - x^T dB y, \tag{C.1}$$

where $L$ is the Lagrange function (4.17).

As explained in section 4.3, $dL$ is independent of $(dx, dy, dX, dY)$ because the variation of the Lagrange function with respect to $(x, y, X, Y)$ vanishes at optimality. The same reasoning implies that the quadratic variation in the objective should be linear in $(dx, dy, dX, dY)$. To compute it, we will work at finite $\mu$. Afterwards, we consider the $\mu \to 0$ limit of the resulting expression and assess the size of finite-$\mu$ corrections.

For brevity, let us write $s = (b, c, B)$ and $z = (x, y, X, Y)$. Given a change $s \to s + ds$, the solution changes as $z \to z + dz + d^2z + \ldots$, where $dz$ and $d^2z$ are linear and quadratic in $ds$, respectively, and "$\ldots$" represent higher order terms in $ds$. The quadratic change in the Lagrange function is

$$d^2 L = \frac{\partial L}{\partial z} d^2 z + \frac{1}{2} \frac{\partial^2 L}{\partial z^2} dz^2 + \frac{\partial L}{\partial s \partial z} ds\, dz + \frac{1}{2} \frac{\partial^2 L}{\partial s^2} ds^2$$
$$= \frac{1}{2} \frac{\partial^2 L}{\partial z^2} dz^2 + \frac{\partial L}{\partial s \partial z} ds\, dz. \tag{C.2}$$

Here, $s$ and $z$ are multidimensional and we suppress indices for brevity. The first term on the first line vanishes by the optimality equations $\frac{\partial L}{\partial z} = 0$, and the last term vanishes because $L$ is linear in $s$. The remaining two terms are proportional to each other. To see this, note that under changing $s \to s + ds$, the shifted optimality equations become

$$0 = \left. \frac{\partial L(s, z)}{\partial z} \right|_{\substack{z \to z + dz + d^2z + \ldots \\ s \to s + ds}}$$
$$= \frac{\partial L(s, z)}{\partial z} + \frac{\partial^2 L(s, z)}{\partial s \partial z} ds + \frac{\partial^2 L(s, z)}{\partial z^2} dz$$
$$= \frac{\partial^2 L(s, z)}{\partial s \partial z} ds + \frac{\partial^2 L(s, z)}{\partial z^2} dz, \tag{C.3}$$

Contracting (C.3) with $dz$ and plugging this result into (C.2), we find

$$d^2 L = \frac{1}{2} \frac{\partial L}{\partial s \partial z} ds\, dz = \frac{1}{2} (db^T dy + dc^T dx - dx^T dB\, y - x^T dB\, dy). \tag{C.4}$$

The variations $dx, dy$ can be computed from the linearized optimality equations (C.3), which are written in more detail in (4.8). After some rearrangement, we find

$$\begin{pmatrix} S & -B \\ B^T & 0 \end{pmatrix} \begin{pmatrix} dx \\ dy \end{pmatrix} = \begin{pmatrix} -dc + dBy \\ db - dB^T x \end{pmatrix}, \tag{C.5}$$

where $S_{pq} = \text{Tr}(A_p X^{-1} A_q Y)$ is the so-called Schur complement matrix. This is precisely the equation solved by SDPB in its main optimization algorithm, with a modified right-hand side. Consequently, it is straightforward to adapt SDPB to determine $dx, dy$ and compute $dL$ and $d^2 L$. We have implemented this computation in a program `approx_objective` packaged with SDPB as of version 2.5.[41]

## C.2 Possible sources of error

We note two possible sources of error in the results for $dL$ and $d^2 L$ — one conceptual and one practical:

$(E_1)$ **Finite-$\mu$ effects.** The formulas for $dL$ and $d^2 L$ were derived assuming finite $\mu$ (so that the optimization problem is well-posed). Is the $\mu \to 0$ limit of these expressions well-behaved? How big are the finite-$\mu$ corrections?

As with the objective function itself, we expect errors in $dL$ and $d^2 L$ to be of order $O(\mu \log \mu)$, provided the SDP is generic. This expectation comes from thinking about $L$ as a function to be optimized $b^T y + c^T x - x^T B y + \text{Tr}\big((X - x^T A_*)Y\big)$, plus a barrier function $-\mu \log \det X$ that imposes that $X$ is positive semidefinite. Near a smooth point on the boundary of the positive-semidefinite cone, the barrier function effectively moves the boundary of the cone by a smoothly-varying amount proportional to $\mu$.

As we vary the parameters $(b, c, B, A)$, the optimal solution with $\mu = 0$ moves along the boundary of the positive semidefinite cone. Similarly, the optimal solution with finite $\mu$ moves along the "effective" boundary a distance $\mu$ away. As long as the boundary is smooth, derivatives of the finite-$\mu$ objective will differ from derivatives of the $\mu = 0$ objective by $O(\mu \log \mu)$ (the size of the barrier function).

$(E_2)$ **Errors from $XY \neq \mu I$.** One of the optimality equations (4.18) is $XY = \mu I$. Under normal operation, SDPB does not attempt to solve this equation with high precision. Instead, it performs repeated Newton steps toward solutions of $XY = \mu^{(i)} I$ with values $\mu^{(i)}$ that *change* with each iteration. This turns the equation $XY = \mu I$ into a kind of moving target. Solutions computed by SDPB will generally have nonzero (but small) $XY - \mu I$.

It is not a-priori obvious how large errors resulting from nonzero $XY - \mu I$ will be. (We show a numerical example in figure 16.) However, they can be mitigated with a simple strategy: After SDPB terminates with a primal-dual optimal solution, we can perform a few extra iterations toward a solution of $XY = \mu I$. In practice, this can be done by running SDPB from the most recent checkpoint with the options listed in table 1 (in addition to whatever other options were used in the optimization). Because the locus $XY = \mu I$ is called the "central path," we call these extra iterations "centering iterations."

---

[41]We thank Walter Landry for collaboration on `approx_objective`.

| option | explanation |
|---|---|
| `--maxIterations=`$n$ | Control the number of iterations. We take $n = 10$ below. |
| `--stepLengthReduction=1` | Take full Newton steps instead of decreasing the step size. |
| `--infeasibleCenteringParameter=1` | Ensure that $\mu$ stays (nearly) constant instead of changing $\mu \to \beta\mu$ with each iteration. This option is only effective if SDPB has both a primal- and dual-infeasible internal state. |
| `--dualityGapThreshold=0` | Ensure a dual-infeasible internal state. |
| `--primalErrorThreshold=0` | Ensure a primal-infeasible internal state. |
| `--dualErrorThreshold=0` | Ensure SDPB doesn't terminate early. |

Table 1: SDPB options for performing centering iterations.

## C.3   Numerical checks

To describe our numerical checks of the expressions for $dL$ and $d^2L$, we need some quick definitions. Given an SDP $s$, let $f(s)$ be the optimal value of its objective. We also define

$$
\begin{aligned}
g(s, ds) &\equiv f(s) + dL + d^2L \\
&= f(s) + \frac{\partial L(s,z)}{\partial s} ds + \frac{1}{2} \frac{\partial L(s,z)}{\partial s \partial z} ds\, dz,
\end{aligned}
\tag{C.6}
$$

where $z$ is the optimum of $s$, and $dz$ (which is linear in $ds$) is the solution to equation (C.3). Note that $g$ is arbitrarily nonlinear in its first argument, but quadratic in its second argument — in fact, $g(s_0, s - s_0)$ provides a quadratic approximation to $f(s)$ around a given $s_0$:[42]

$$
f(s) = g(s_0, s - s_0) + O((s - s_0)^3).
\tag{C.7}
$$

Consider now a family of SDP's $s(\Delta)$ depending smoothly on a parameter $\Delta$. Consider a sequence of values $\Delta_0 + \delta\Delta$ converging to $\Delta_0$, and let us write $s_0 = s(\Delta_0)$. Equation (C.7) with $s = s(\Delta_0 + \delta\Delta)$ implies that

$$
h(\delta\Delta) \equiv f(s(\Delta_0 + \delta\Delta)) - g(s_0, s(\Delta_0 + \delta\Delta) - s_0) \sim O(\delta\Delta^3),
\tag{C.8}
$$

where we used that $s(\Delta)$ depends locally smoothly on $\Delta$. We can use this to check our expressions for $dL$ and $d^2L$: we compute $h(\delta\Delta)$ for several values of $\delta\Delta$ and check whether it decreases cubically in $\delta\Delta$.

---

[42]In other words, $g$ is a 2-jet of $f$ at $s_0$.

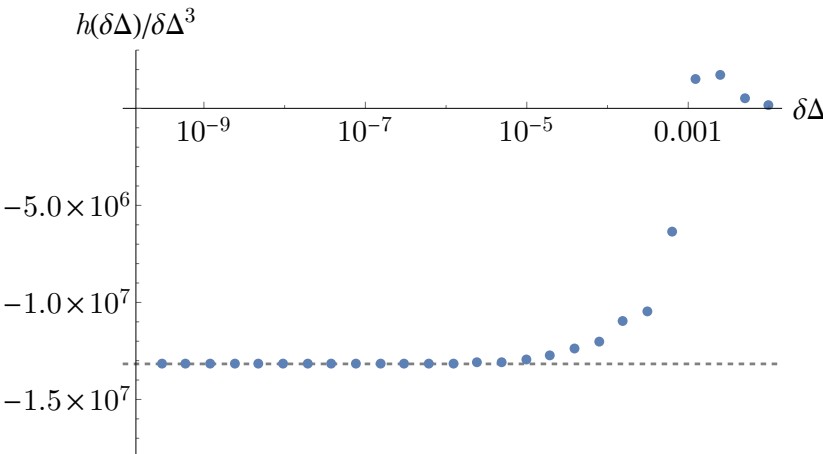

Figure 15: The ratio $h(\delta\Delta)/\delta\Delta^3$ for a family of SDPs describing $\sigma$ and $\epsilon$ correlators in the 3d Ising model. Specifically, we studied the GFF navigator function in the 2-parameter 3d Ising setup described in section 2.2, with fixed $\Delta_\epsilon = 1.4$ and varying $\Delta_\sigma = 0.518 + \delta\Delta$, where $\delta\Delta = 0.01 \times 2^{-n}$ and $n \in \{0, \dots, 25\}$, and derivative order $\Lambda = 11$. We see that the difference between the true objective and its quadratic approximation is cubic in $\delta\Delta$. The optimizations for this plot were computed with a duality gap threshold of $10^{-30}$, and 10 centering iterations.

In figure (15), we plot the ratio $h(\delta\Delta)/\delta\Delta^3$ for a one-parameter family of SDP's describing the GFF navigator function for correlators of $\sigma$ and $\epsilon$ in the 3d Ising model. For small $\delta\Delta$, the ratio $h(\delta\Delta)/\delta\Delta^3$ approaches a constant. This is a strong check of our results for $dL$ and $d^2L$ and our ability to compute them accurately: cubic dependence of $h(\delta\Delta)$ on $\delta\Delta$ requires delicate cancellations between the true objectives of $s(\Delta_0 + \delta\Delta)$ and $s_0$, the linear correction $dL$, and the quadratic correction $d^2L$. The SDPB computations in figure 15 were performed with duality gap threshold $D = 10^{-30}$, with 10 centering iterations. Evidently these choices effectively remove both sources of error $(E_1)$ and $(E_2)$ in this example.[43]

In figures 16 and 17, we show the effects of $(E_1)$ and $(E_2)$ on $dL$ and $d^2L$. Figure 16 was produced with no centering iterations, so it shows the effects of both $(E_1)$ and $(E_2)$. In that case, the relative error in $dL$ scales approximately as $\mu^{0.8}$, and the relative error in $d^2L$ scales as $\mu^{0.235}$. These numbers presumably are not universal: they depend on the whole history of the optimization procedure in SDPB, and are not uniquely determined by the final solution. Figure 17 was produced with 10 centering iterations. In that case, the errors in $dL$ and $d^2L$ both scale linearly with $\mu$, and are much smaller overall. This is strong evidence that centering iterations effectively mitigate $(E_2)$, and it also supports our estimate of the size of finite-$\mu$ effects.

---

[43]More precisely $(E_1)$ and $(E_2)$ are unimportant for the values of $\delta\Delta$ shown in the plot. They will become important again at smaller values of $\delta\Delta$. To get accurate results for even smaller $\delta\Delta$, we can decrease $\mu$ by further lowering the duality gap threshold.

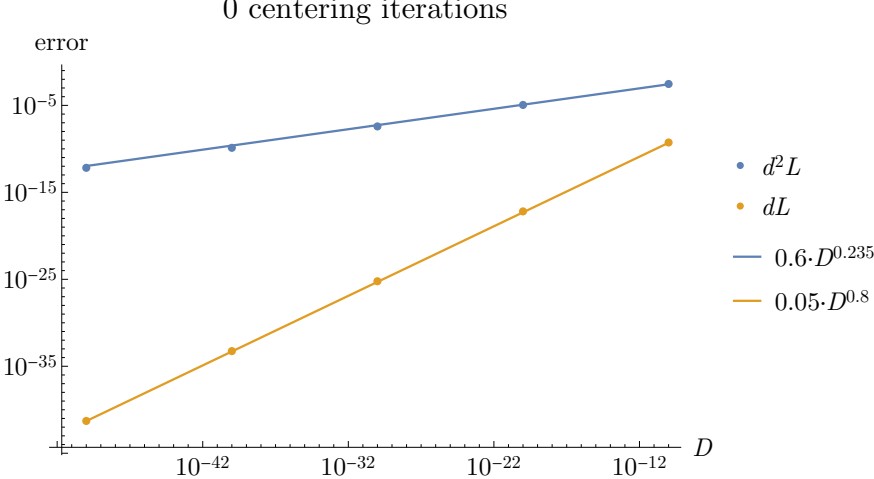

Figure 16: The relative error in $dL$ and $d^2L$, as a function of the duality gap $D$ (which is proportional to $\mu$), computed with no centering iterations. We use the setup described in the caption of figure 15, with $\delta\Delta = 0.01 \times 2^{-25}$. We define relative error for a quantity $x$ by $|x - x_{\text{ref}}|/|x_{\text{ref}}|$, where $x_{\text{ref}}$ is a reference value. Reference values for this plot were computed with duality gap $10^{-50}$ and 30 centering iterations. For both $dL$ and $d^2L$, we show best fits to powers of $D$.

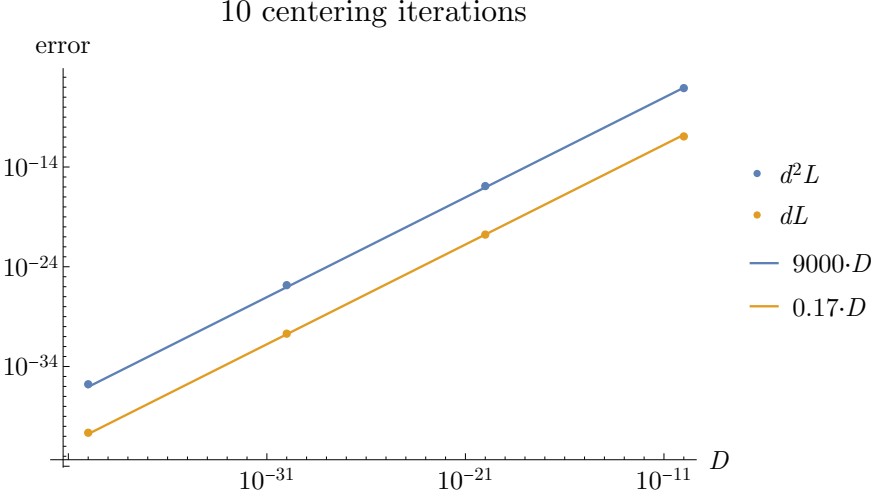

Figure 17: Errors for $dL$ and $d^2L$ as a function of the duality gap $D$ with the same setup as figure 17, but where for each optimization we perform 10 centering iterations of SDPB. The errors now decrease linearly with $D$ (which is proportional to $\mu$). This is consistent with our naive estimate $\mu \log \mu$ in section C.2. (To detect the logarithm $\log \mu$, we would need more data and a more careful fit.)

| Section(s) | 3 | 3 | 5.3, 6 | 5.3 |
|---|---|---|---|---|
| $\Lambda$ | 11 | 19 | 11 | 19 (PyCFTBoot, see below) |
| keptPoleOrder[45] | 8 | 14 | 14 | |
| order | 60 | 60 | 27 | |
| spins | $\{0,\ldots,21\}$ | $\{0,\ldots,26,49,50\}$ | $\{0,\ldots,27\}$ | $\{0,\ldots,28\}$ |
| precision | 640 | 768 | 768 | 660 |
| dualityGapThreshold | $10^{-30}$ | $10^{-30}$ | $10^{-20}$ | $10^{-30}$ |
| primalErrorThreshold | $10^{-30}$ | $10^{-30}$ | $10^{-60}$ | $10^{-30}$ |
| dualErrorThreshold | $10^{-30}$ | $10^{-30}$ | $10^{-60}$ | $10^{-30}$ |
| initialMatrixScalePrimal | $10^{20}$ | $10^{40}$ | $10^{20}$ | $10^{20}$ |
| initialMatrixScaleDual | $10^{20}$ | $10^{40}$ | $10^{20}$ | $10^{20}$ |
| feasibleCenteringParameter | 0.1 | 0.1 | 0.1 | 0.1 |
| infeasibleCenteringParameter | 0.3 | 0.3 | 0.3 | 0.3 |
| stepLengthReduction | 0.7 | 0.7 | 0.7 | 0.7 |
| maxComplementarity | $10^{100}$ | $10^{100}$ | $10^{100}$ | $10^{100}$ |

Table 2: Parameters used to setup the SDPs, along with the SDPB parameters. The definition of these can be found in [5] (where order was 90 and keptPoleOrder was $\kappa$).

# D    Parameters for numerics

The computation of the navigator function can be translated to the form of a semidefinite program (SDP), to solve which we use the arbitrary precision solver SDPB [5, 6]. We used simpleboot [29], PyCFTBoot [30], and sdpb-haskell[44] to setup the SDPs. The parameters used for the computations are presented in Table 2. We used the same conformal block normalization as [1].

For the $\Lambda = 19$ results in Section 5.3, we used the Python package PyCFTBoot [30] to setup the SDP, with parameters $(k_{max}, l_{max}, n_{max}, m_{max}) = (28, 28, 1, 9)$. The parameters $(n_{max}, m_{max})$ control the number of derivatives used in the $(a, b)$ coordinates (see [30] for more details). This choice results in the same navigator value as taking $(z, \bar{z})$ derivatives up to $\Lambda = 19$.

To numerically implement the BFGS Algorithm 1, we have used the BFGS algorithm minimize(method='BFGS') of Python's SciPy library, with the additional modifications of the rescaling of the initial Hessian and the implementation of the bounding box. All parameters used were the default ones, both for the Moré and Thuente line search SciPy implements and the actual BFGS algorithm.

# E    Further plots

Here we collect plots like Figs. 7 and 8 for six additional runs of our modified BFGS algorithm, for both the two parameter $\Lambda = 11$ case discussed in Sec. 5.3.1, and the

---

[44]https://gitlab.com/davidsd/sdpb-haskell

[45]The computations presented in Sections 5.3 and 6 were set up using a version of simpleboot where the definition of keptPoleOrder was slightly different. Here the poles were kept without modifying the residue to better approximate the contribution of discarded poles and thus the blocks were less accurate than those used in [5].

three parameter $\Lambda = 19$ case discussed in Sec. 5.3.2

## E.1   2-parameter searches

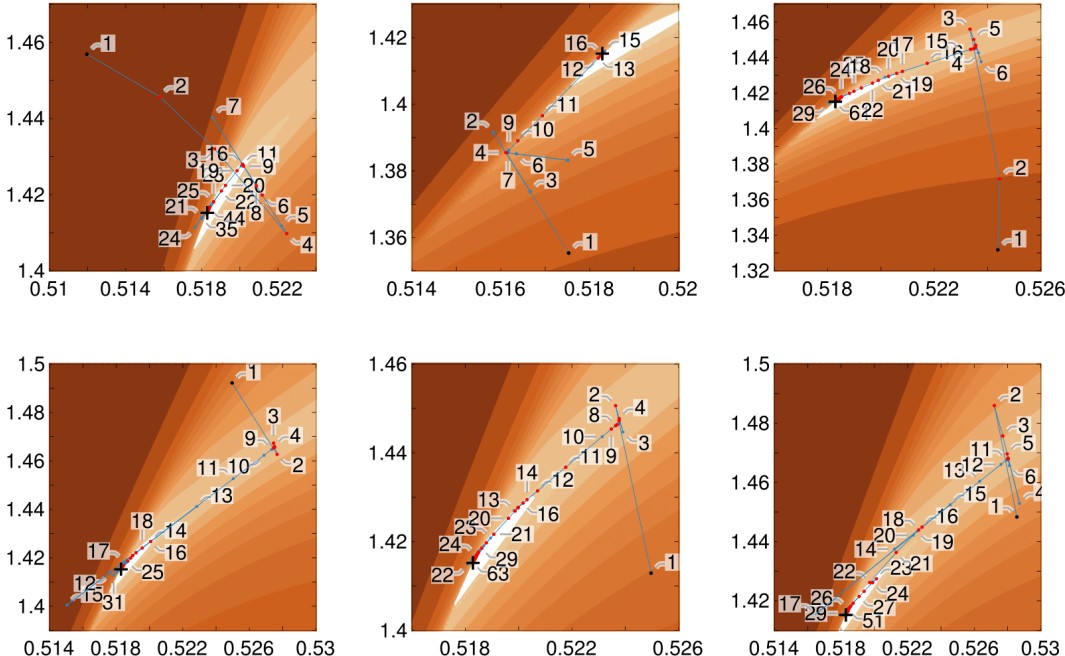

Figure 18:   Six more runs of our algorithm, see Section 5.3.1, in addition to the run shown in Fig. 7. Plotting conventions are the same as in that figure.

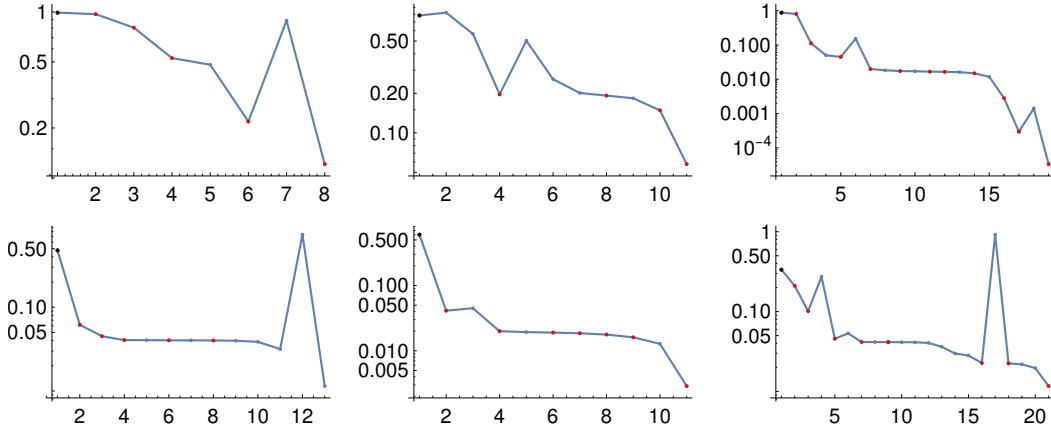

Figure 19: This plot is analogous to Fig. 8(left). It shows navigator values $\mathcal{N}_i$ at the $i$-th function call for the 6 runs from Fig. 18, and with the same color code for the dots.

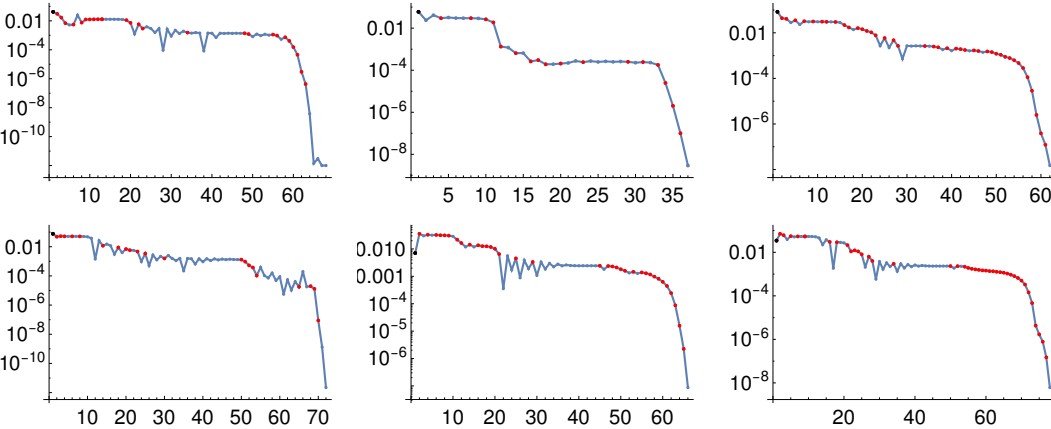

Figure 20: This plot is analogous to Fig. 8(right). It shows logarithmic plots of $\|x_i - x_f\|$ at the $i$-th function call for the 6 runs from Fig. 18, and with the same color code for the dots.

## E.2  3-parameter searches

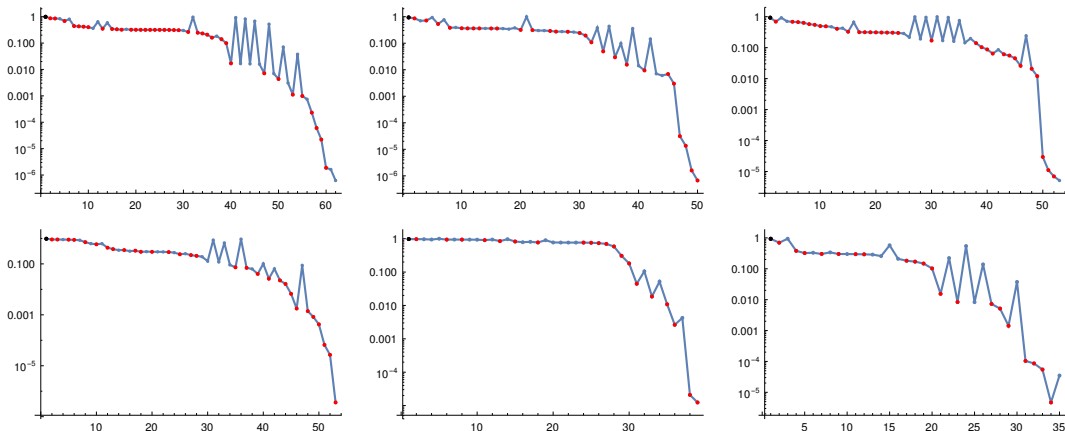

Figure 21: Same as Fig. 12(left), for 6 additional runs appearing in Fig. 10. The figure shows navigator values $\mathcal{N}_i$ at the $i$-th function call for the 6 additional runs, with the same color code for the dots as in Fig. 12 .

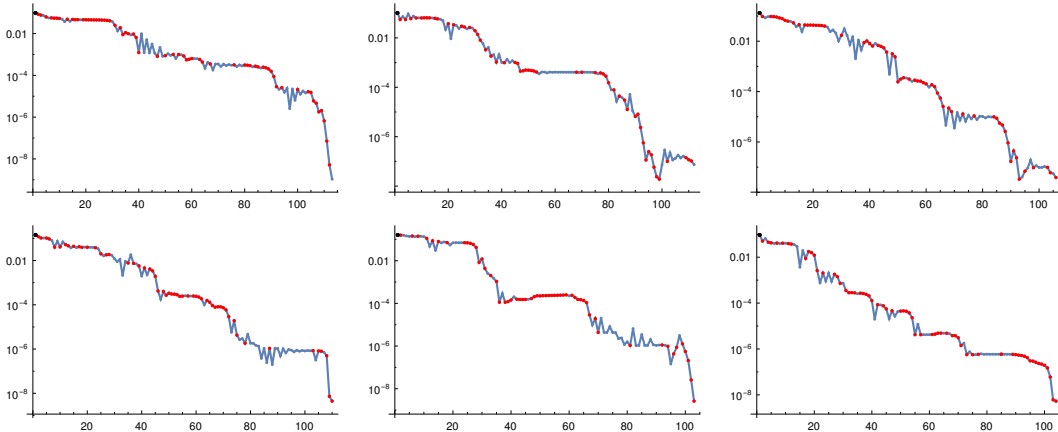

Figure 22: Same as Fig. 12(right), for 6 additional runs appearing in Fig. 10. The figure shows logarithmic plots of $\|x_i - x_f\|$ at the $i$-th function call for the 6 additional runs, with the same color code for the dots as in Fig. 12.

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
