# Peer review of "Navigator Function for the Conformal Bootstrap"

_SciPost Physics_

## Round 1 · Referee Report · Anonymous (Referee 1) · 2021-5-26

Strengths

  1. Presents a very useful new algorithm to more efficiently explore the space of CFTs using the conformal bootstrap.

  2. Should have wide applicability in the numerical bootstrap field.

Weaknesses

  1. The content of the paper is very technical and might not be of interest for the wider physics community outside of hardcore numerical bootstrappers.

Report

This paper presents navigator functions that can be used to efficiently map out the space of the allowed region in the conformal bootstrap. The key property of this function is its minima correspond to allowed region, and its computationally cheap to compute its gradient, which makes it efficient to find allowed regions. The paper gives examples of the practical use of this navigator function for the Ising model.

Requested changes

  1. Most of the calculations were done with the GFF navigator, and the authors claim that the Sigma navigator gives similar results. The Sigma navigator has a more general definition than the GFF navigator, as it depends on some specific subset of operators. It would be useful to give more intuition of which subsets of operators give the best nagivator (say for the Ising model), and in general how these Sigma nagivators compare to GFF in terms of efficiency and accuracy.

  2. The authors mention that the minimum of the navigator likely corresponds to a CFT. Does that mean for conformal manifolds, one would find a flat plane?

  3. This paper focuses on islands in the space of scaling dimensions (combined with ratios of ope coefficients in some cases). One can also compute islands purely in the space of OPE coefficients for operators whose scaling dimensions are known a priori (such as the stress tensor if one puts a gap above it, or protected operators in supersymmetric theories), where OPE extremization immediately allows one to find the boundary of the island. As the authors briefly remark, the gradient formula for the navigator function would work in a similar way for OPE extremization, it would be useful to comment on how this could be used to more efficient compute such OPE space islands.

---

## Round 1 · Referee Report · Anonymous (Referee 2) · 2021-6-14

Strengths

  1. This paper introduces the idea of a navigator function into the numerical conformal bootstrap. This will be useful in future bootstrap studies where one needs a systematic method to find allowed points in CFT parameter space and to efficiently extremize various parameters.

  2. The paper contains a number of results which make it easier to implement the navigator function idea in practice. It gives three concrete proposals for navigator functions along with a detailed study of the BFGS algorithm (and some modifications) applied to one of the navigator functions. It also usefully explains how gradient and Hessian information can be readily computed using the output of the semidefinite program encoding the crossing relations.

Weaknesses

  1. I don't see any major weaknesses.

Report

This is a nicely written paper which opens a new pathway in conformal bootstrap research. I would recommend publication in SciPost after a few minor improvements described below.

Requested changes

  1. In the abstract it is stated that the navigator functions introduced in the paper are everywhere well-defined. This doesn't seem to be completely true, e.g. the GFF navigator (2.6) only seems to be well-defined in the region $2\Delta_{\phi} < \Delta_{*} \leq 2\Delta_{\phi} + 2$. So, I would suggest that the language be moderated to reflect this.

  2. In a number of places in the paper, the number of function calls (or steps) are described. E.g., in section 6.2, 10 steps of the navigator algorithm are compared with 480 points tested in the Delaunay search approach. However, it would be more useful to compare the total number of SDPB iterations taken across these 10 steps (presumably involving O(100) iterations each), compared to the total number of SDPB iterations in the 480 points of the Delaunay search (presumably involving O(1-10) iterations each. I would like to see the authors add a brief discussion of how the SDPB iterations compare between the navigator method and Delaunay triangulation method.

  3. It would be helpful if the authors could add a comment or plot showing the value of the navigator function inside the Ising island. E.g., how flat is the function across the island? How does the value of the navigator at its minima (5.8, 5.9) compare to the value at the best estimates for the Ising model?

---

## Round 1 · Referee Report · Kay Joerg Wiese (Referee 3) · 2021-6-16

Strengths

  • 2 new conceptual results
  • worked out examples
  • enormous potential for speed up of the numerical bootstrap.

Weaknesses

-in parts rather technical

Report

This work provides two key results for the numerical bootstrap:
(1) it constructs a "navigator function", positive outside the allowed region, and negative inside. It is clear that this additional information has an enormous potential for an increase in speed
(2) a formula for the derivatives of the navigator function, obtained essentially for free, and allowing one to use faster minimisation algorithms.
Result (2) is obtained in two different ways: the second derivation, via a variational principle, is enlightening as it shows that this relation, and potentially more, are to be expected.
The ideas are then tested on the 3d Ising model, and shown to perform well. Potential pitfalls are: local minima, which do not seem to exist, or non-convex regions, which slow down the applied search algorithms. A partial solution to the latter is presented.

The work is written in the style of a PhD thesis, with many details and worked-out examples. This has its advantages, but makes the reading in parts lengthy. Due to the importance of the tools proposed, I recommend leaving the choice with the authors.

Requested changes

figs 12 and 13: colours mentioned in the caption not visible on my screen.

---

## Editorial Decision

resubmitted